# A deep siamese neural network improves metagenome-assembled genomes in microbiome datasets across different environments

Shaojun Pan [1,2], Chengkai Zhu[1,2,3], Xing-Ming Zhao [1,2,4,5✉] & Luis Pedro Coelho [1,2✉]

Metagenomic binning is the step in building metagenome-assembled genomes (MAGs) when sequences predicted to originate from the same genome are automatically grouped together. The most widely-used methods for binning are reference-independent, operating de novo and enable the recovery of genomes from previously unsampled clades. However, they do not leverage the knowledge in existing databases. Here, we introduce SemiBin, an open source tool that uses deep siamese neural networks to implement a semi-supervised approach, i.e. SemiBin exploits the information in reference genomes, while retaining the capability of reconstructing high-quality bins that are outside the reference dataset. Using simulated and real microbiome datasets from several different habitats from GMGCv1 (Global Microbial Gene Catalog), including the human gut, non-human guts, and environmental habitats (ocean and soil), we show that SemiBin outperforms existing state-of-the-art binning methods. In particular, compared to other methods, SemiBin returns more high-quality bins with larger taxonomic diversity, including more distinct genera and species.

[1] Institute of Science and Technology for Brain-Inspired Intelligence, Fudan University, Shanghai, China. [2] Key Laboratory of Computational Neuroscience and Brain-Inspired Intelligence, Ministry of Education, Shanghai, China. [3] School of Life Sciences, Fudan University, Shanghai, China. [4] MOE Frontiers Center for Brain Science, Fudan University, Shanghai, China. [5] Zhangjiang Fudan International Innovation Center, Shanghai, China. ✉email: xmzhao@fudan.edu.cn; luispedro@big-data-biology.org

Metagenomic sequencing is commonly used to study microorganisms in the environment without a need for culturing[1–4]. Computational analysis of metagenomic data has enabled the construction of large compendia of metagenome-assembled genomes (MAGs) from human-associated[5–7], animal-associated[8], and environmental[9,10] samples. These have been used to discover thousands of new human microbiome species spanning different human body sites[5,7], provide a genomic catalog from several habitats[11], and reconstruct thousands of nucleocytoplasmic large DNA viruses[12].

The standard approach, which we follow here, is to assemble quality-controlled short reads into longer contigs, followed by binning, i.e., grouping the contigs into "bins" such that each bin is predicted to contain only contigs from the same genome. An alternative pipeline consists of building a gene catalog[13], followed by grouping the genes in the catalog using co-abundance[14]. This results in collections of genes representing a metagenomic species and methods have been extended to recover both core and accessory genes[15]. In this work, we focus on binning contigs which have been assembled from short reads.

Binning methods can be divided into two categories, depending on whether they rely on pre-existing genomes (reference-dependent) or operate de novo (reference-independent). Reference-dependent methods are limited to discovering novel genomes from previously-known species, while reference-independent ones can recover completely novel species and even novel phyla[16,17].

Most reference-independent methods are completely unsupervised, such as Canopy[14], Metabat2[18], Maxbin2[19], VAMB[20], and COCACOLA[21], relying on sequence features (e.g., $k$-mer frequencies) and co-abundance (coverage of the reads mapping to the contig[14]), which are assumed to vary more between than within genomes. One exception is SolidBin[22], which uses a semi-supervised approach[23]. In particular, SolidBin takes advantage of must-link and cannot-link constraints between pairs of contigs: a must-link constraint indicates that two contigs should be binned together, while a cannot-link one indicates that they should not (despite their name, these constraints are not strictly followed by the algorithm). SolidBin generates these constraints by alignment to existing genomes, marking pairs of contigs that align to the same species as must-link, while pairs that align to different genera are considered as cannot-link. Generating these constraints by alignment to reference genomes is, however, subject to both noise (from annotation error) and sampling bias in genome databases (see Supplementary Text).

Here, we introduce SemiBin (Semi-supervised metagenomic Binning), a binning method based on contrastive learning with deep siamese neural networks to take advantage of must-link and cannot-link constraints[24] (see Fig. 1). We first used simulated data from the Critical Assessment of Metagenome Interpretation (CAMI)[25,26], to evaluate SemiBin in a setting where the expected genomes are known. Additionally, we used metagenomes from four different habitats (human gut, dog gut, marine, and soil microbiomes) where we relied on automated tools to estimate the quality of the results. Although the downstream use of the bins may determine the level of quality that is optimal for each particular study, as SemiBin can be generically applied, we focused on the number of high-quality bins that may be considered MAGs[27]. Overall, SemiBin outperformed other tools in all conditions tested.

## Results

**SemiBin is a binning tool based on deep learning**. SemiBin is a tool for metagenomic binning at contig level which uses deep contrastive learning (see Fig. 1, Supplementary Figs. 1 and 2). It relies on the use of must-link and cannot-link constraints between contigs. Cannot-link constraints are derived from contig annotations to the Genome Taxonomy Database (GTDB)[28] as we observed that this could be performed robustly (see Supplementary Text and Supplementary Table 2). Must-link constraints, however, are generated by breaking up longer contigs artificially. For each contig, SemiBin computes $k$-mer frequencies or $k$-mer frequencies and abundance (depending on the number of samples used, see Methods). A deep siamese neural network[24] (i.e., a network consisting of two identical subnetworks, so that pairs of inputs are processed in parallel) is then used to transform the original $k$-mer and abundance features into a transformed space (the embedded features, see Methods). SemiBin then generates a sparse graph from these embedded features and groups the contigs into bins with the Infomap[29] community detection algorithm. After that, SemiBin reclusters bins whose number of single-copy genes is greater than one using the weighted $k$-means algorithm to get the final binning results (see Methods).

Studies often collect multiple related metagenomes, and there have been different proposals on how to handle binning in this context: single-sample (where each metagenome is handled independently), co-assembly (where different metagenomes are pooled together), or multi-sample (where resulting bins are sample-specific, but abundance information is aggregated across samples)[20]. All three modes are supported by SemiBin.

**Semi-supervised learning improves binning results**. To show the impact of semi-supervised learning in SemiBin, we compared the proposed method to the same pipeline without the deep learning feature embedding step, an approach we termed as NoSemi (see Methods). We evaluated the performance on five simulated datasets from CAMI I and CAMI II[25,26] (the Critical Assessment of Metagenome Interpretation). The CAMI I datasets comprise different numbers of organisms including strain variation with low (40 genomes, 1 sample), medium (132 genomes, 2 samples), and high (596 genomes, 5 samples) complexity and were used to evaluate single-sample (low complexity) and co-assembly (medium and high complexity) binning. CAMI II datasets mimic different human body sites and contain multiple simulated metagenomes from the same environments, thus allowing us to test multi-sample binning.

SemiBin could reconstruct 6.7–65.0% more distinct, high-quality bins compared to this non-embedding version in the five datasets, and the improvement was especially pronounced in complex environments (see Supplementary Figs. 3 and 4).

To evaluate the learning ability of the siamese neural network, namely that it has the ability to learn the underlying genome structure from the must-link and cannot-link constraints, not just reproduce its inputs, we compared the full pipeline to NoSemi (no constraints are used) as well as SemiBin_m, SemiBin_c, and SemiBin_mc which directly use the must-link and cannot-link constraints to generate the graph without the semi-supervised learning step (see Methods). The complete SemiBin pipeline performed similarly or better (average 12.4% more high-quality bins) in the low and medium complexity datasets and showed large improvements (average 34.6% more high-quality bins) in the high complexity dataset (see Supplementary Fig. 5) compared to versions that directly use the must-link and cannot-link constraints. SemiBin could also reconstruct an average 7.0% and 16.0% more distinct high-quality strains in Skin and Oral datasets, respectively (see Supplementary Fig. 4).

These results showed that the siamese neural network had the ability to learn the underlying structure of the environment from the must-link and cannot-link constraints and could improve the binning results.

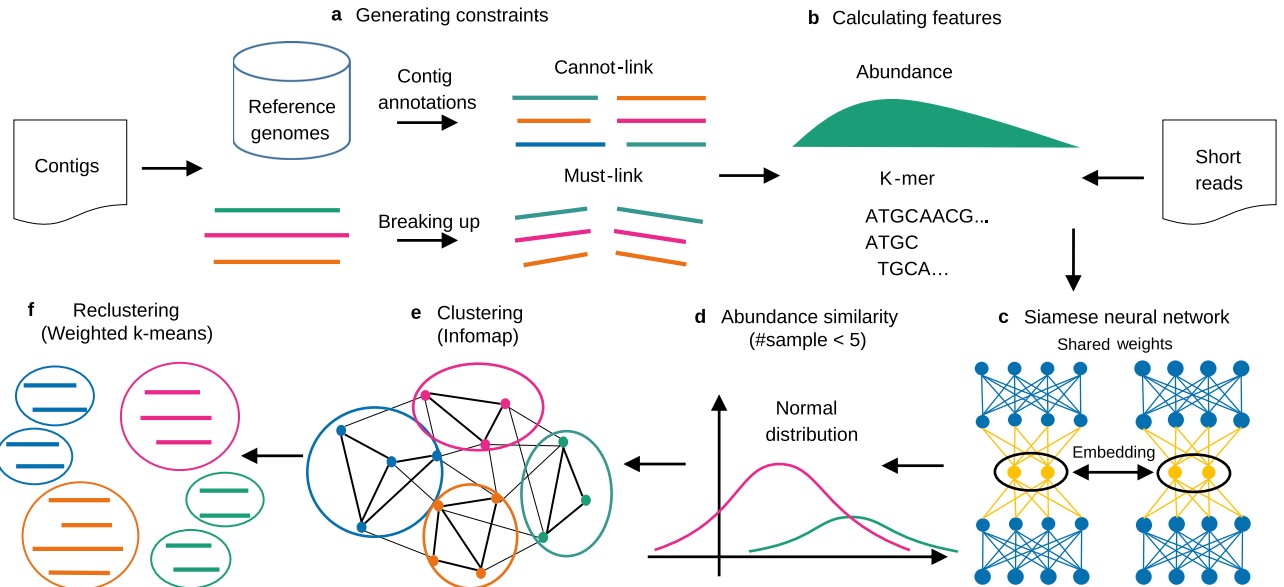

**Fig. 1 Overview of the SemiBin pipeline. a** generate must-link constraints by breaking up contigs artificially and cannot-link constraints based on contig taxonomic annotations (i.e. GTDB reference genomes). **b** calculate abundance estimates (average and variance of the number of reads per base) and *k*-mer frequency of every contig. **c** train siamese neural network using the cannot-link and must-link constraints as inputs (*k*-mer frequencies and abundance features). The learned embedding will be the features from the output layer of the neural network and be used in step **e** for binning. **d** Based on the assumption that the number of reads per base obeys a normal distribution, calculate the Kullback-Leibler divergence of the normal distributions of two contigs. SemiBin uses this value as the abundance similarity when the number of samples used is smaller than 5. **e** generate a sparse graph with the embedded distance and abundance similarity between contigs and uses the Infomap algorithm to obtain the preliminary bins. **f** SemiBin uses weighted *k*-means to recluster contigs in the bins whose mean number of single-copy genes is greater than one to get the final bins.

**SemiBin outperforms existing binners in simulated data sets.** To compare SemiBin to existing binners, we benchmarked SemiBin, Metabat2[18], Maxbin2[19], VAMB[20], SolidBin[22], and COCACOLA[21] on five simulated datasets from CAMI I and CAMI II[25,26] (see Methods) with single-sample, co-assembly, and multi-sample binning (see Methods).

In the CAMI I datasets, SemiBin was able to reconstruct the highest number of high-quality bins (completeness >90% and contamination <5%[27]) and achieved the best F1-score (see Fig. 2a, Supplementary Figs. 6 and 7). SemiBin reconstructed 17.4%, 11.4% and 11.4% more high-quality bins compared to the second-highest result in the low, medium, and high complexity datasets respectively. Since metagenomic binning is particularly challenging when multiple strains from the same species are present[30], we evaluated the performance of the methods on common strains (defined as genomes for which another genome with at least 95% average nucleotide identity (ANI) is present[25]) in the three datasets. SemiBin was able to reconstruct 11.1%, 18.5%, and 35.3% more high-quality common strains than the second-best alternative (see Supplementary Fig. 7).

In the CAMI II datasets, we compared SemiBin to VAMB and Metabat2 on the Skin (610 genomes, 10 samples) and Oral (799 genomes, 10 samples) datasets, as previous benchmarking studies had indicated their best performance on these[20]. We benchmarked the original Metabat2 and an adapted version that implements multi-sample binning (see Methods). This adaptation, however, led only to modest improvements (see Supplementary Fig. 8a). Compared to VAMB (the second-best binner), SemiBin could reconstruct 44.3% and 24.8% more distinct high-quality strains, 34.9% and 22.7% more distinct species, and 23.7% and 14.3% more distinct genera in the Skin and Oral datasets respectively (see Fig. 2b). We evaluated the behavior of the methods on multi-strain species and observed that SemiBin could reconstruct more high-quality distinct strains across almost all ANI intervals, even when very similar genomes (ANI >99.5%) are present (see Supplementary Fig. 9).

**SemiBin outperforms existing binners in real datasets.** To test SemiBin on real data, we applied it to datasets from four different environments: human gut ($n = 82$)[31], non-human animal-associated (dog gut, $n = 129$)[32], ocean surface samples from the Tara Oceans project[33] ($n = 109$), and soil[34] ($n = 101$). In large-scale metagenomic analyses, single-sample binning is widely used (see Supplementary Table 3) because samples can be trivially processed in parallel. Here, we compared SemiBin to Maxbin2, VAMB, and Metabat2 with single-sample binning. Additionally, because Nissen et al.[20] have demonstrated the effectiveness of multi-sample binning in real projects, we compared SemiBin with VAMB in this mode.

Because the actual genomes are unknown in real data, we relied on CheckM[35] and GUNC[36] to evaluate the quality of the recovered bins (see Methods). In all cases, SemiBin recovered more high-quality bins than the alternatives considered (see Fig. 3b and Supplementary Fig. 10). In the human gut, dog gut, ocean, and soil datasets, SemiBin reconstructed 1497, 2415, 446, and 95 high-quality bins with single-sample binning, significantly outperforming Metabat2 with an increase of 437 (41.2%), 1011 (72.0%), 146 (48.7%), and 36 (61.0%), respectively. For multi-sample binning, SemiBin could reconstruct 17.5%, 11.0%, 30.7%, and 171.4% more high-quality bins than VAMB. In the human gut, ocean, and soil datasets, SemiBin with single-sample binning performed better than VAMB with multi-sample binning, which requires more computational time for short-read mapping and cannot be performed in parallel. We annotated these high-quality bins from multi-sample binning with GTDB-Tk[37]. SemiBin could reconstruct more distinct taxa when compared to VAMB in the human gut, ocean, and soil datasets, performing similarly in the dog gut dataset (see Supplementary Fig. 11).

Generating the cannot-link constraints and training the models comes at a substantial computational cost. For the human gut, an average sample required ca. 276 minutes on a CPU (a graphical unit, GPU, can reduce the time to 129 minutes), with RAM usage

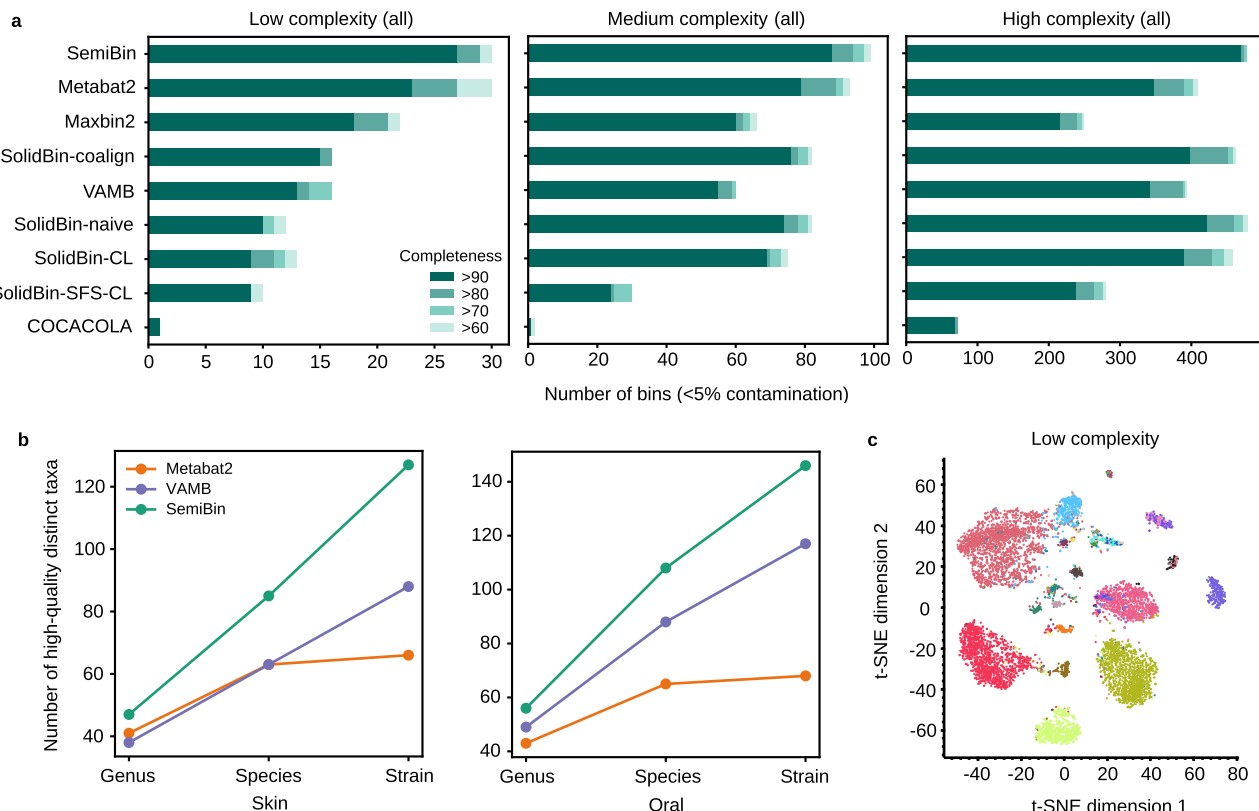

**Fig. 2 SemiBin outperformed other binners in simulated datasets with single-sample, co-assembly, and multi-sample binning. a** In CAMI I simulated datasets, SemiBin returned more high-quality bins. Shown are the numbers of reconstructed genomes per method with varying completeness and contamination <5% (methods shown, top to bottom: SemiBin, Metabat2, Maxbin2, SolidBin-coalign, VAMB, SolidBin-naive, SolidBin-CL, SolidBin-SFS-CL, and COCACOLA). **b** SemiBin reconstructed a larger number of distinct high-quality genera, species, and strains in the CAMI II Skin and Oral datasets compared to either Metabat2 or VAMB. A high-quality strain is considered to have been reconstructed if any bin contains the strain with completeness >90% and contamination <5% (see Methods). If at least one high-quality strain is reconstructed for a particular genus or species, then those are considered to have been reconstructed. **c** Semi-supervised embedding separates contigs from different genomes. Shown is a two-dimensional visualization of embedding of the low complexity dataset from CAMI I, with contigs colored by their original genome (using t-SNE, as implemented in scikit-learn, parameters: perplexity = 50, init = "pca"[77]). Source data are provided as a Source Data file.

peaking at 38 GB (see Supplementary Tables 5 and 6). By default, SemiBin learns a new embedding model for each sample, which can require a large computational expenditure for large projects. Therefore, for single-sample binning, the possibility of reusing a learned model from one sample to another (model transfer) was tested with encouraging results (see Supplementary Fig. 12 and Supplementary Text). However, there was still a loss in the number of high-quality bins recovered compared to learning a new model for every sample. To overcome this, we built models from multiple samples, an approach we termed SemiBin(pretrain). Using a pretrained model, again in the human gut, reduces the per-sample time to 7.2 minutes and peak memory to 4.4 GB (see Supplementary Tables 5 and 6).

Furthermore, if a large enough number of samples is used in pretraining (see Methods), SemiBin(pretrain) can reconstruct more high-quality bins and outperformed Metabat2 in all cases (see Fig. 3a). In the human gut and dog gut datasets, SemiBin(pretrain) trained on more than three samples outperformed the original SemiBin and performed at a similar level in the ocean and soil datasets. As pretraining is performed only once and applying the model to other samples is computationally fast, SemiBin(pretrain) can be used in large-scale metagenomic analyses (see Supplementary Tables 5 and 6).

For every environment, we chose the pretrained model that performed best to include in the benchmark (for human gut, dog gut and ocean datasets, a model trained from 20 samples;

for soil dataset, a model trained from 15 samples; see Fig. 3). Compared to the original SemiBin, SemiBin(pretrain) could reconstruct 203 (13.6%), 382 (15.8%), and 8 (8.4%) more high-quality bins in human, dog gut and soil datasets and achieved similar results in the ocean dataset. When compared to Metabat2, SemiBin(pretrain) could reconstruct 640 (60.4%), 1393 (99.2%), 144 (48.0%) and 44 (74.6%) more high-quality bins. SemiBin (pretrain) also performed significantly better than Metabat2 when comparing high-quality bins on a sample-by-sample basis (Wilcoxon signed-rank test, two-sided, $P = 3.687 \times 10^{-15}$ $(n = 82)$, $P = 5.987 \times 10^{-23}$ $(n = 129)$, $P = 7.232 \times 10^{-12}$ $(n = 109)$, $P = 4.434 \times 10^{-07}$ $(n = 101)$; see Fig. 3b).

We used Mash[38] to identify instances when SemiBin(pretrain) and Metabat2 reconstructed the same genome. Most Metabat2-generated high-quality bins corresponded to high-quality bins generated by SemiBin(pretrain). SemiBin(pretrain) results further contained many high-quality bins which corresponded to lower-quality (or absent) bins in the Metabat2 results, the inverse case (Metabat2 generating a higher-quality version of a SemiBin(pre-train) bin) being relatively rare. For genomes that were recovered at high-quality by both binners, the SemiBin(pretrain) bins had, on average, higher completeness and F1-score, with only a minimal increase in contamination in the human gut, dog gut, and ocean datasets (not statistically significant in the human and dog gut datasets, see Fig. 4 and Supplementary Fig. 13).

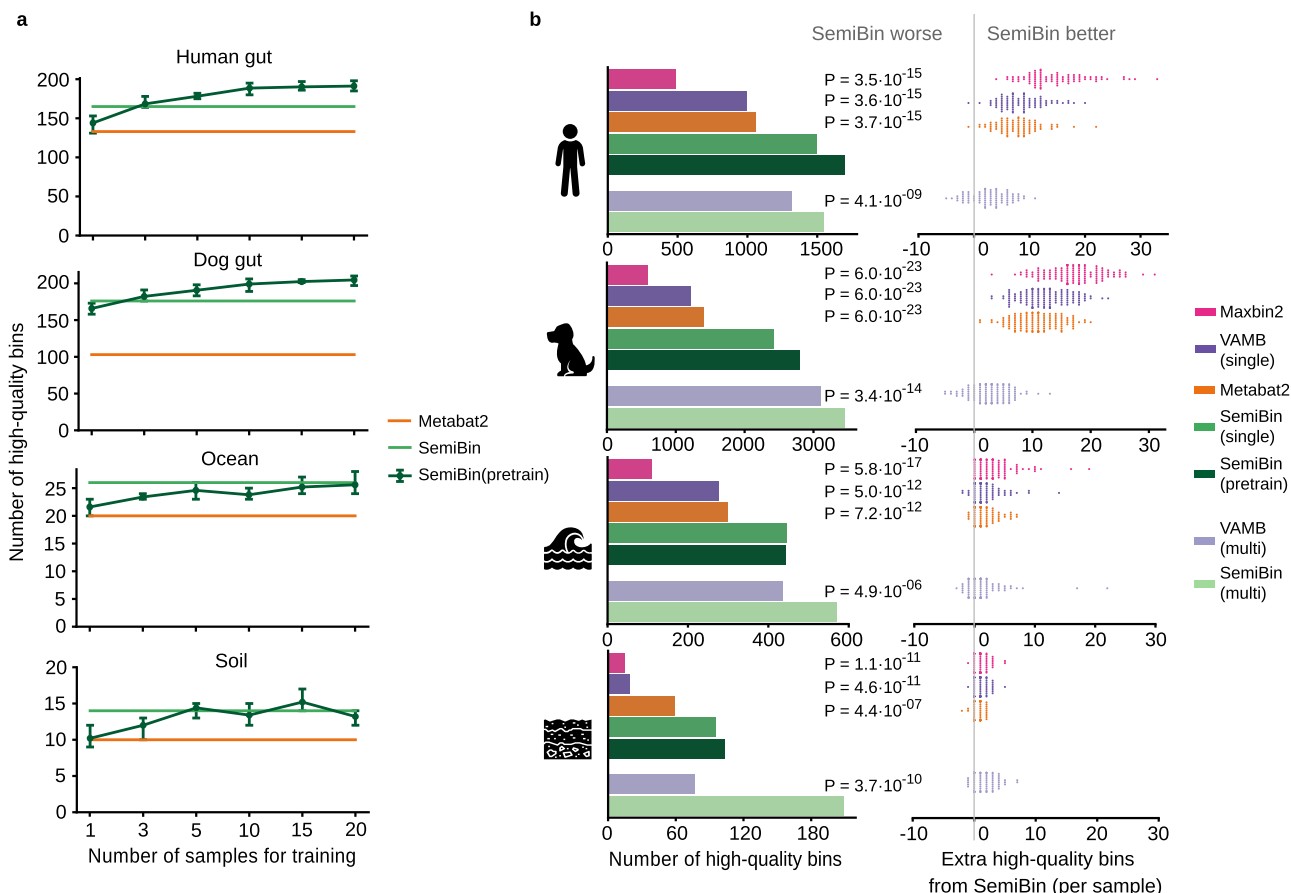

**Fig. 3 SemiBin outperformed other binners in real datasets. a** Number of high-quality bins (automatically evaluated using CheckM and GUNC, see Methods) using a pretrained model with different numbers of samples (per-sample SemiBin and Metabat2 shown for comparison). Data are presented as mean and range (the maximum datapoint minus the minimum) from the five runs. **b** SemiBin(pretrain) produced more high-quality bins compared to Maxbin2, VAMB and Metabat2 in the human gut ($n = 82$ samples), dog gut ($n = 129$ samples), ocean ($n = 109$ samples) and soil ($n = 101$ samples) data sets (left). When considering the results in each sample (right), SemiBin(pretrain) outperformed other single-sample binners in almost every sample. In the dog gut dataset, SemiBin(pretrain) always produced more high-quality bins than any other per-sample method. In the human gut, ocean, and soil data sets, other methods occasionally outperformed SemiBin(pretrain) (observed for two human gut samples, nine ocean samples, and five soil samples), but the difference is never large (at most, two extra high-quality bins were produced). Results of VAMB in multi-sample binning mode are compared to SemiBin(multi), with SemiBin(multi) producing more high-quality bins overall (although not in every sample). P-values shown are from a Wilcoxon signed-rank test (two-sided) on the counts for each sample. Source data are provided as a Source Data file.

We annotated the high-quality bins generated by SemiBin(pretrain) and Metabat2 with GTDB-Tk[37]. Bins from SemiBin(pretrain) represented a larger taxonomic diversity at all levels (see Supplementary Fig. 14). Additionally, SemiBin(pretrain) was able to recover more genomes from both known and unknown species (see Supplementary Fig. 15), validating the ability of the deep learning model to extract information from the background data while being capable of going beyond it.

To further test the generalization ability of the learned model, we applied the pretrained model from the human gut dataset to two human gut datasets not used for model training (hold-out datasets), a setting we termed SemiBin(pretrain; external). As the human gut dataset considered so far is from a German population (where individuals are assumed to consume a Western-style diet), we tested both on another dataset from a German population[39], and on a dataset from a non-Westernized African human population[7]. SemiBin(pretrain; external) significantly outperformed Metabat2 (Wilcoxon signed-rank test, two-sided, $P = 3.050 \times 10^{-12}$ ($n = 92$), $P = 1.014 \times 10^{-7}$ ($n = 50$); see Supplementary Fig. 16), showing that the pretrained model could be used on hold-out data sets.

Besides these four environments, we also provide pretrained model for six other environments (cat gut, human oral, mouse gut,

pig gut, built environment, and wastewater; the environments from GMGCv1[10] which contain enough deeply sequenced samples for building a model). For every environment, except wastewater (both tools extract very few bins from this environment, see Fig. 5), SemiBin(pretrain) could generate 11.8–240.4% more high-quality bins than Metabat2 (see Fig. 5). We also measured the effects of model transfer between environments. In most of the environments, the pretrained model from the same environment returned the best results (with only small differences otherwise). Even transferring from different environments, however, Semi-Bin(pretrain) was still able to outperform Metabat2 in most cases.

Finally, we trained a model using samples from all environments and termed it as SemiBin(global) (see Supplementary Text). SemiBin(global) outperformed Metabat2 in all environments. Compared to SemiBin(pretrain), SemiBin(global) performed similarly or slightly worse in most of the environments (SemiBin(pretrain) much better in the human and dog gut environments, see Fig. 5).

**SemiBin can discover differences in *Bacteriodes vulgatus* strains from human and dog gut samples.** Amongst species shared between the human and dog gut microbiomes in our

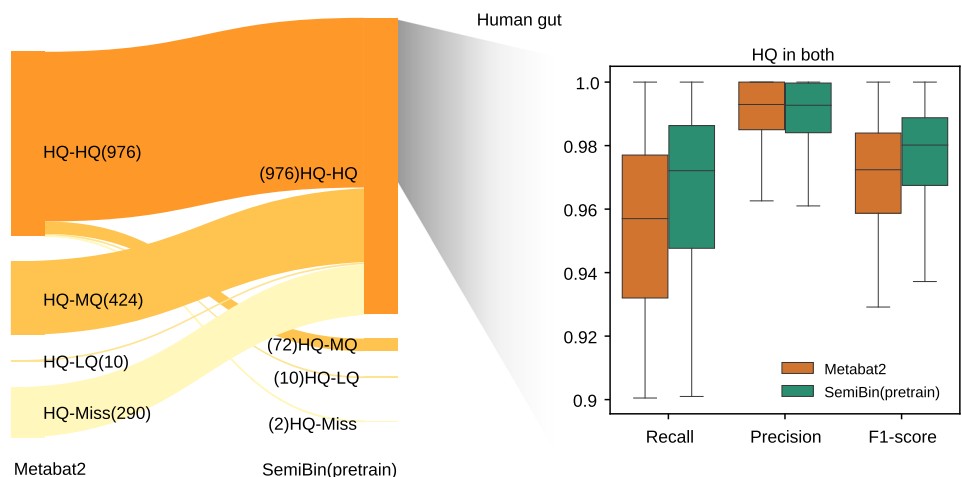

**Fig. 4 SemiBin(pretrain) reconstructed more and better high-quality bins compared to Metabat2 in human gut dataset.** We identified the overlap between the bins in the human gut from Metabat2 and SemiBin(pretrain) using Mash (see Methods). Within bins that are present at high-quality in the output of both binners, SemiBin(pretrain) achieved higher completeness (recall) ($P = 2.236 \times 10^{-67}$ ($n = 82$)) without a statistically significant increase in contamination ($1-$precision) ($P = 0.167$ ($n = 82$)). The overall F1-score is significantly better ($P = 9.287 \times 10^{-67}$ ($n = 82$); all $P$ values were computed using the Wilcoxon signed-rank test, two-sided null hypothesis). Similar results were obtained in the dog gut and ocean datasets (see Supplementary Fig. 13). For the box plots, the centerline is the median of all values, the lower and upper bounds of the box correspond to 25th and 75th percentiles and the lower and upper of the whiskers are the minimum and maximum values. Source data are provided as a Source Data file.

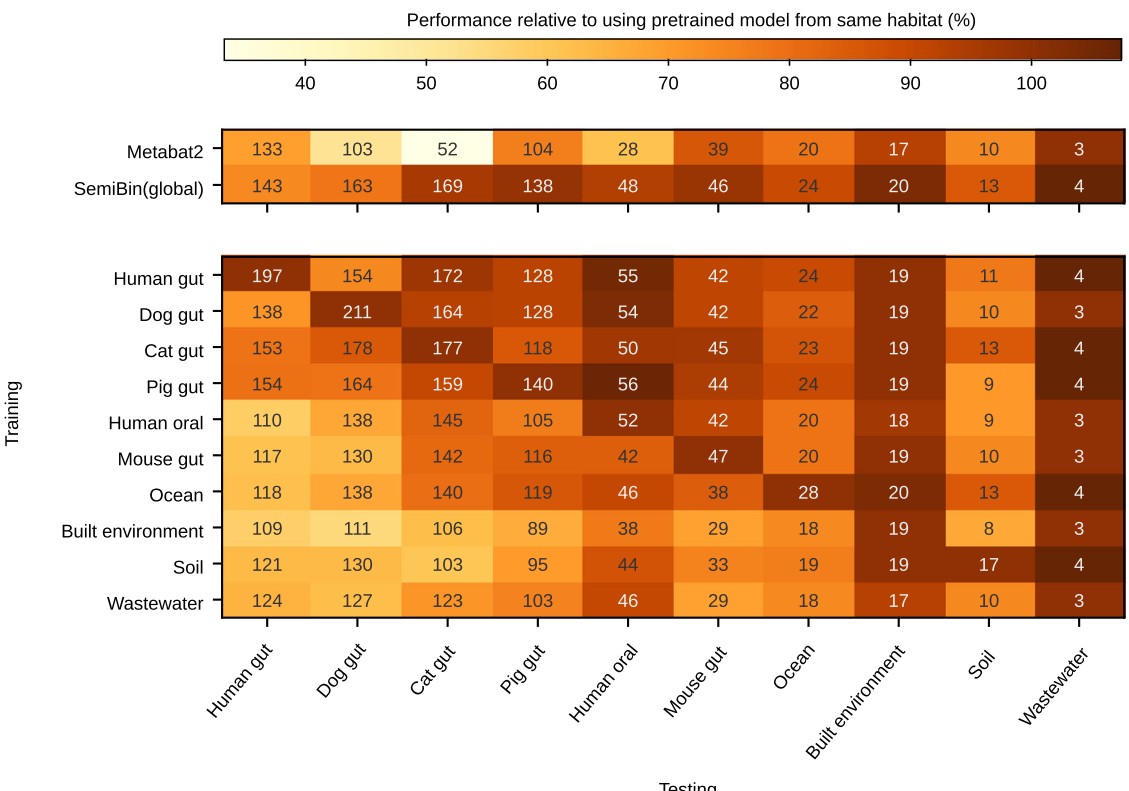

**Fig. 5 Habitat-specific models outperform SemiBin(global) which outperforms Metabat2 across 10 environments.** We tested every model on the 10 environments (10 testing samples, no overlap with the training samples). We also trained a model from all environments (training from all samples used to generate the pre-trained model for the 10 environments). We termed this model as SemiBin(global). Shown in each cell is the number of high-quality bins obtained from the testing samples, while the color indicates the performance relative to using the model trained in the same environment (a pseudo-count of 10 was added to the raw numbers to smooth the estimates). In most environments, the pre-trained model from the same environment as the testing environment returns the best results. When transferring a model to a different environment, the model can also get good results, and in most situations, it still performs better than Metabat2. In particular, in dog gut, cat gut, and human oral, SemiBin(pretrain) performed better than Metabat2 when training from any environment.

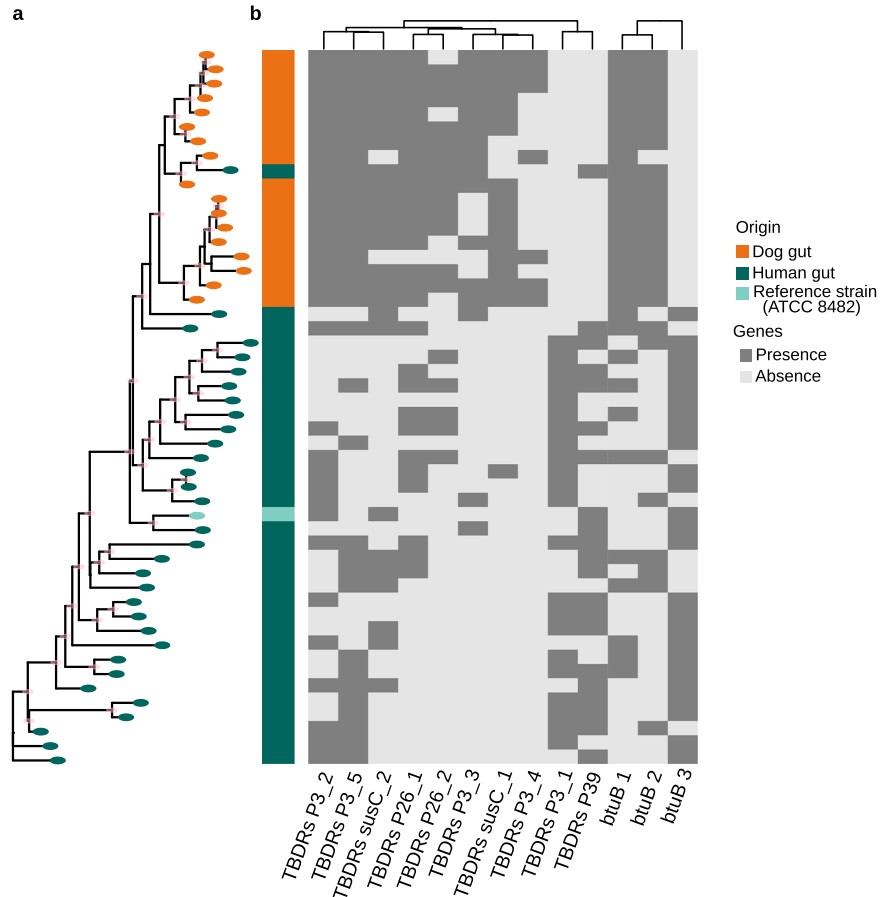

**Fig. 6 Strains of *B. vulgatus* recovered from the human and dog gut microbiomes and the type strain *B. vulgatus* ATCC 8482 cluster according to their host. a** Shown is the maximum-likelihood phylogenetic tree based on core genes; branches with bootstrap values higher than 70 were marked pink in the nodes. Clustering based on ANI or gene presence also showed a separation between the hosts (see Supplementary Fig. 17). **b** The gene content of the strains was also statistically different between the two hosts (*P* < 0.05, *n* = 50, using Fisher's exact test after FDR correction using the Benjamini–Hochberg method, see Methods). In particular, the presence of different TonB-dependent receptors (TBDR) and their interaction partner *btuB*, which are involved in nutrient uptake, differs (see Supplementary Text). Source data are provided as a Source Data file.

study, *B. vulgatus* had the largest number of recovered strains in both. Namely, 17 dog gut and 32 human gut *B. vulgatus* bins were obtained with SemiBin(pretrain) (compared to four and five obtained with Metabat2). Based on either sequence similarity or gene content, the bins from the dog gut clustered separately from those of the human gut (see Fig. 6, Supplementary Fig. 17). These also correspond to different gene presence patterns for genes encoding TonB-dependent receptors, ten of which showed statistically significant differences (after multiple hypothesis correction; see Fig. 6, Supplementary Data 1). We also found that the three *btuB* genes in dog and human gut microbiomes (see Fig. 6 and Supplementary Data 1) encoding a protein that interacts with TonB[40] belong to different orthologous groups. As different btuB-TonB complexes may affect the transport of essential micronutrients such as vitamin B12[41], *B. vulgatus* in human and dog gut microbiomes might have different degrees of ability to transport vitamin B12.

## Discussion

Compared to SolidBin, an existing semi-supervised binning tool, SemiBin uses a better way to generate must-link constraints, namely breaking up contigs artificially. This is an approach that fits into the framework of self-supervised learning as it exploits the structure of the problem without the need for external

labels[42]. As chimeric contigs are rare[43], this results in a very low rate of must-link false positives. In the future, we will explore the possibility of extending self-supervision to other components of the pipeline.

Cannot-link constraints are generated in a reference-dependent manner using taxonomic annotation of contigs. We explored two tools for taxonomic annotation: CAT[44] (using the NCBI taxonomy) and MMseqs2[45,46] (using the GTDB). Interestingly, on simulated data, using CAT resulted in more high-quality bins recovered (although the difference was small); on real data, however, using MMseqs2 returned better results (see Supplementary Text and Supplementary Fig. 18). This illustrates the perils of over-reliance on simulated benchmarks, which do not capture all the complexity of real data.

One limitation of the default SemiBin approach of training a new model per sample is that generating contig annotation and training the model is computationally costly (see Supplementary Table 5). However, we also showed that when multiple samples are available, training a model from a subset of samples and applying it to the remaining ones can result in more high-quality genomes at minimal marginal computational cost per sample.

Models learned from samples from the same habitat are recommended, although transferring between habitats still results in a good performance. Generally speaking, transferring between similar habitats[10], such as between the guts of different mammals,

resulted in better performance than transfers between dissimilar ones. Learning a generic model resulted in a loss of performance compared to a specialized model, but as it still outperformed Metabat2, we recommend its use for applying SemiBin to new habitats if training a habitat-specific model is not feasible (e.g., too few samples are available or the computational costs are too high).

In summary, we showed superior results of SemiBin when compared to other state-of-the-art binning methods, demonstrating the advantages of using background knowledge in metagenomic binning. Looking forward, we expect that reference genome databases will continue to improve in quality along with taxonomic prediction methods. This should make the use of this information even more valuable and further widen the gap with respect to purely de novo methods.

## Methods

**SemiBin pipeline**. SemiBin is developed with Python. The input to SemiBin consists of a set of contigs and mapped short reads (in the form of BAM files) from one or multiple samples. The output consists of a collection of bins with the original contigs clustered.

*Preprocessing*. Prior to binning, contigs are filtered to remove short contigs, with the default minimal length for binning being 2500 bp. If the contigs in the range 1000–2500 bp represent <5% of the total number of basepairs (this parameter can be changed by the user), then the threshold is reduced to 1000 bp.

After filtering, every contig is represented by $k$-mer relative frequencies **K** and abundance distribution **A**. SemiBin uses $k=4$ (also used in other tools[18–22]), so the dimension of **K** is 136 ($k$-mers that are the reverse complement of each other are considered equivalent). $K$-mer vectors are normalized by the lengths of each contig after the addition of a small pseudo-count, namely:

$$k'_i = \frac{k_i + 10^{-5}}{\sum_{j=1}^{136} k_j + 10^{-5}} \tag{1}$$

The abundance $a$ is defined as the average number of reads per base mapping to the contig. We calculated the abundance with BEDTools (version 2.29.1, genomecov command)[47]. If the number of samples $N$ used in the binning is greater than or equal to 5, SemiBin scales the abundance features from the $N$ samples to a similar order of magnitude as the $k$-mer frequencies (which are restricted to the 0–1 range, by construction). In particular, the constant $s$ used to scale was defined as $100 \times \lceil \frac{a_{mean}}{100} \rceil$. After the processing, the input to the semi-supervised model is the vector $\mathbf{Z} = [k'_1, k'_2 \ldots k'_{136}; \frac{a_1}{s_1}, \frac{a_2}{s_2} \ldots \frac{a_N}{s_N}]$. Otherwise, when the number of samples used is smaller than 5, the input to the semi-supervised model is the vector $\mathbf{Z} = [k'_1, k'_2 \ldots k'_{136}]$.

*Generating must-link and cannot-link constraints*. SemiBin uses taxonomic annotation results to generate cannot-link constraints and breaks up contigs to build must-link constraints. By default, MMseqs2[45,46] (version 13.45111, default parameters) is used to annotate contigs with GTDB[28] reference genomes. We defined contigs pairs with different annotations at species level (with scores both above 0.95) or at genus level (with scores both above 0.80) as containing a cannot-link constraint between them. If the number of cannot-link constraints is too large (default is four million), they are randomly sampled to speed up the training process. SemiBin also employs the same method as Maxbin2[19] with FragGeneScan (version 1.30)[48] and HMMER (version 3.1.b1)[49] to predict seed contigs for different species using single-copy marker genes and generates cannot-link constraints between them.

To generate must-link constraints, SemiBin breaks up long contigs into two fragments with equal length artificially and generates must-link constraints between the two shorter contigs. By default, SemiBin uses a heuristic method to automatically determine the minimum size of contigs to break up (alternatively, the user can specify the threshold): SemiBin selects a minimum size that makes contigs as long as or longer than this threshold can contain most (98%) basepairs of the input contigs. If this minimum size is smaller than 4000 bp, the minimum will be set to 4000 bp.

*Semi-supervised deep siamese neural network architecture*. We used a semi-supervised siamese neural network (see Supplementary Figs. 1 and 2)[24] for dealing with the must-link and cannot-link constraints. The network has two siamese neural networks with shared weights.

As it is semi-supervised, this neural network is trained with two loss functions. The supervised loss is a contrastive loss that is used to classify pairs of inputs as must-link or cannot-link (in the contrastive loss function, we termed the must-link

constraints as positive labels and cannot-link constraints as negative labels):

$$L^{Con} = \frac{1}{|\mathbf{M_x} \cup \mathbf{C_x}|} \sum_{(x_1, x_2) \in |\mathbf{M_x} \cup \mathbf{C_x}|} y d(x_1, x_2)^2 \\ + (1 - y) \max\left\{1 - d(x_1, x_2), 0\right\}^2 \tag{2}$$

where $\mathbf{M_x}$ denotes the must-link pairs and $\mathbf{C_x}$ the cannot-link pairs in the training set, $d(x_1, x_2)$ is the Euclidean distance of the embedding of $(x_1, x_2)$, and $y$ is an indicator variable for the $(x_1, x_2)$ pair (with the value 1 if $(x_1, x_2) \in \mathbf{M_x}$, and 0 if $(x_1, x_2) \in \mathbf{C_x}$).

The goal of the supervised embedding is to transform the input space so that contigs have a smaller distance if they are in the same genome, compared to pairs of contigs from different genomes. To ensure that the embedding learns structure is shared by all genomes, we also used an autoencoder[50] to reconstruct the original input from the embedding representation with an unsupervised mean square error (MSE) loss function:

$$L^{MSE} = \frac{1}{|\mathbf{X}|} \sum_{x \in \mathbf{X}} (x - \hat{x})^2 \tag{3}$$

where $x$ is the original input, $\hat{x}$ is the reconstructed input, and $\mathbf{X}$ are all contigs in the dataset.

The model used here is a dense deep neural network (see Supplementary Fig. 2). The dimension of the input features, $F$ depends on the number of samples used in the binning (see "Preprocessing" Section). The first two layers of the encoding and the decoding network are followed by batch normalization[51], a leaky rectified linear unit[52], and a dropout layer[53] (the dropout rate used was 0.2). The purpose of the training is to optimize the contrastive loss and the MSE loss at the same time with the Adam[54] optimization algorithm. The model was implemented in Pytorch[55].

*Similarity between contigs*. The similarity between two contigs is defined as

$$S(x_1, x_2) = 1 - \begin{cases} \min(d(\hat{x}_1, \hat{x}_2), 1) & \text{if } \geq 5 \text{ samples} \\ \min(d(\hat{x}_1, \hat{x}_2) \cdot a(x_1, x_2), 1) & \text{otherwise;} \end{cases} \tag{4}$$

where $d(\hat{x}_1, \hat{x}_2)$ is the Euclidean distance of the semi-supervised embedding. The embedding of the contig is the features (100 dimensions) from the output layer of the encoder of the semi-supervised siamese neural network (see Supplementary Fig. 19). When there are fewer than five samples, the embedding distance only contains $k$-mer information. In this case, we modeled the number of reads per base of one contig as a normal distribution[56], and used the Kullback-Leibler divergence[57] to measure the divergence between the normal distributions from contigs, denoted as $a(x_1, x_2)$ above.

*Clustering the contigs*. The binning problem is modeled as clustering on a graph. First, SemiBin considers the fully connected graph with contigs as nodes and similarity between contigs as the weight of edges. To convert the community detection task to an easier task, the fully connected graph is converted into a sparse graph. A parameter (max_edges, defaulting to 200) is used to control the sparsity of the graph. For each node, only the max_edges edges with the highest weights are kept. To remove any potentially artefactual edges introduced by the embedding procedure, SemiBin builds another graph with the same procedure using the original features ($k$-mer frequencies and abundance features). The edges in the graph built from embedding that does not exist in the graph from original features are also removed.

After building the sparse graph, Infomap (python-igraph package[58], version 0.9.7), an information-theory-based algorithm to reveal community structure in weighted graphs[29], is used to reconstruct bins from the graph. If the user requests it, SemiBin can use single-copy genes of the reconstructed bins to independently re-bin bins whose mean number of single-copy genes is greater than one[19]. For this, SemiBin uses the weighted $k$-means algorithm to recluster bins according to the embedded and the abundance features. Finally, SemiBin outputs the final binning results, removing bins smaller than a user-definable threshold (which defaults to 200 kbp).

*SemiBin(pretrain)*. The default pipeline of SemiBin is to (1) generate must-link/cannot-link constraints, (2) train the siamese neural network model for this sample, (3) bin based on the embeddings. To address the issue that contig annotations and model training requires significant computational time and considering that the trained models can be transferred between samples or even projects, we propose to (1) train a model with constraints from one sample or several samples and (2) apply this model to other samples without further learning. This approach is termed SemiBin(pretrain). To use SemiBin(pretrain) in the tool, users can train a model from their datasets or use one of our 10 built-in pretrained models.

*Binning modes*. We have evaluated SemiBin in three binning modes: single-sample, co-assembly, and multi-sample binning[20]. Single-sample binning means binning each sample into inferred genomes after independent assembly. This mode allows for parallel binning of samples, but it does not use information across samples.

Co-assembly binning means that samples are co-assembled first and then binning contigs with abundance information across samples. This mode can generate longer contigs and use co-abundance information, but co-assembly may

lead to inter-sample chimeric contigs[1] and binning based on co-assembly cannot retain sample-specific variation[20].

Multi-sample binning means that the resulting bins are sample-specific (as in single-sample binning), but the information is aggregated across samples (in our case, abundance information). This mode requires more computational resources as it requires mapping reads back to a database consisting of contigs from all the samples.

In single-sample and co-assembly binning, we calculate the $k$-mer frequencies and abundance for every sample and bin contigs from every sample independently. For multi-sample binning, we first concatenate contigs from every sample into a single database to which reads are mapped to generate abundance profiles. Unlike what is used in VAMB (which introduced the multi-sample binning concept), this concatenated database is then re-split and contigs from each sample are binned separately.

**Data used**. For the benchmarking of binners, we used five simulated datasets from the CAMI challenges (CAMI I and CAMI II) and six real metagenomic datasets. Five simulated datasets from CAMI I and CAMI II were downloaded from the CAMI challenge[25,26]. CAMI I includes three datasets: low complexity, medium complexity, and high complexity datasets. The low complexity dataset has 40 genomes with a single simulated sample. The medium complexity dataset has 132 genomes with two simulated samples with different insert sizes. Here, we used samples with 150 bp insert size. The high complexity dataset has 596 genomes with 5 samples. We also used the skin and oral cavity datasets from the toy Human Microbiome Project data in CAMI II. The Skin dataset contains 10 samples with 610 genomes, while the Oral dataset contains 10 samples with 799 genomes. We used the low complexity dataset to evaluate the single-sample binning mode of our method, the medium and high complexity datasets to evaluate the co-assembly binning mode, and the Skin and Oral datasets to evaluate the multi-sample binning mode. We used fastANI[59] (version 1.32, default parameters) to calculate the ANI value between genomes for every sample from the CAMI II datasets.

We also used six real microbiome projects from different environments:

1. a German human gut dataset with 82 samples[31] (study accession PRJEB27928),
2. a dog gut dataset with 129 samples[32] (study accession PRJEB20308),
3. a marine dataset from the Tara Oceans project with 109 ocean surface samples[33] (study accessions PRJEB1787, PRJEB1788, PRJEB4352, and PRJEB4419),
4. a soil dataset with 101 samples[34] (for accessions, see Supplementary Data 2),
5. a second German human gut dataset with 92 samples[39] (study accession PRJNA290729),
6. a non-Westernized African human gut dataset with 50 samples[7] (study accession PRJNA504891).

We used the first four datasets to evaluate single-sample and multi-sample binning mode and the last two human gut projects as hold-out datasets to evaluate transferring with a pretrained model in SemiBin. We also used samples of cat gut ($n = 30$)[60], human oral ($n = 30$)[61], mouse gut ($n = 30$)[62], pig gut ($n = 30$)[63], built environment ($n = 30$)[64] and wastewater ($n = 17$)[65] environments from GMGCv1[10] to generate pretrained models. There are four other habitats in GMGCv1, but they did not contain enough deeply sequenced samples for training and testing. For the human oral dataset, we removed repeated samples from the same individual to ensure that samples are independent. For every environment, we used 20 samples for training the model (except for the case of wastewater, where seven were used) and 10 samples for testing the results (*no* overlap between the training and testing samples). We additionally trained a model from all environments (training from all samples used to generate the pretrained model for the 10 environments considered here). We termed this model SemiBin(global).

For simulated datasets, we used the gold standard contigs provided as part of the challenge. For real datasets (except PRJNA504891), short reads were first filtered with NGLess[66] (version 1.0.1, default parameters) to remove low-quality reads and host-matching reads (human reads for human gut datasets, dog reads for dog gut datasets). These preprocessed reads were then assembled using Megahit[67] (version 1.2.4, default parameters) to assemble reads to contigs. For PRJNA504891, we used Megahit[67] (version 1.2.8, default parameters) to assemble reads to contigs. We mapped reads to the contigs with Bowtie2[68] (version 2.4.1, default parameters) to generate the BAM files used in the binning. For multi-sample binning mode, contigs from all samples were collated together into a single FASTA file. Then reads from every sample were mapped to the concatenated database to obtain a BAM file for each sample.

**Methods included in the benchmarking**. We compared SemiBin to other methods in three binning modes. For single-sample and co-assembly binning of CAMI I datasets, we compared our method to the following methods: Maxbin2 (version 2.2.6)[19], Metabat2 (version 2)[18], VAMB (version 3.0.2)[20], COCACOLA (git version 707e284a74b9a9257bec6dfe08205939a210ea31)[21], SolidBin (version 1.3)[22]. SolidBin is the only existing semi-supervised binning method and it has different modes. Here, we focused on the comparison to modes that use information from reference genomes: SolidBin-coalign (which generates must-link constraints from reference genomes), SolidBin-CL (which generates cannot-link constraints from reference genomes), SolidBin-SFS-CL (which generates must-link constraints from feature similarity and reference genomes). We also added SolidBin-naive (without additional information) to show the influence of different semi-supervised modes.

For multi-sample binning of CAMI II datasets, we compared to the existing multi-sample binning tool VAMB, which clusters concatenated contigs based on co-abundance across samples and then splits the clusters according to the original samples and default Metabat2. For more comprehensive benchmarking, we converted Metabat2 to multi-sample mode. We used jgi_summarize_bam_contig_depths (with default parameters) to calculate depth values using the BAM files from every sample mapped to the concatenated contig database. Then, we ran Metabat2 to bin contigs for every sample with abundance information across samples, which is a similar idea to the multi-sample mode in SemiBin. This adaptation, however, led only to modest improvements (see Supplementary Fig. 8a)

We also benchmarked single-sample and multi-sample binning modes in real datasets. For single-sample binning, we compared the performance of SemiBin to Maxbin2, Metabat2, and VAMB; and, for multi-sample binning, we compared to VAMB. These tools have been shown to perform well in real metagenomic projects (see Supplementary Table 3). For VAMB, we set the minimum contig length to 2000 bp. For SolidBin, we ran SolidBin with constraints generated from annotation results with MMseqs2[45,46] and we used the binning results after postprocessing with CheckM. For SemiBin, we ran the whole pipeline described in the Methods (with default parameters). For other methods, we ran tools with default parameters.

To evaluate the effectiveness of the semi-supervised learning in SemiBin, we also benchmarked modified versions of SemiBin:

1. NoSemi, where we removed the semi-supervised learning step in the SemiBin pipeline, and clustered based on the original inputs.
2. SemiBin_m, where we removed the semi-supervised learning step, but directly used the must-link constraints in the sparse graph (by adding a link between must-link contigs).
3. SemiBin_c was analogous to SemiBin_m, but we used cannot-link constraints by removing links in the sparse graph between cannot-link contigs.
4. SemiBin_mc combines the operations of SemiBin_m and SemiBin_c.

By default, SemiBin is trained on each sample to obtain the embeddings that are used for extracting the final bins. For SemiBin(pretrain), we trained the model from several samples and applied the pretrained model to the whole project (see Fig. 3). We also used two hold-out projects to show the generalization of the pretrained model (see Supplementary Fig. 16). For other benchmarking which used 10 samples as the testing set to evaluate the pretrained model, there was no overlap between the training and testing samples (see Figs. 3a, 5 and Supplementary Fig. 12).

**Computational resource usage**. For evaluating computational resource usage in a standardized condition, we used two Amazon Web Services (AWS) virtual machines. We used the machine type g4ad.4xlarge with 1 CPU (2nd generation AMD EPYC processors) containing 8 physical cores, 16 logical cores, and 64 GB RAM memory to run Maxbin2, Metabat2, VAMB, and SemiBin in CPU-mode. Additionally, we used the type g4dn.4xlarge to run VAMB and SemiBin in GPU-mode. This machine contains an NVIDIA Tesla T4 GPU.

**Evaluation metrics**. For simulated datasets in CAMI, we used AMBER (version 2.0.1)[69] to calculate the completeness (recall), purity (precision), and F1-score to evaluate the performance of different methods.

In the real datasets, as ground truth is not available, we evaluated the completeness and contamination of the predicted bins with CheckM[35] (version 1.1.3, using lineage_wf workflow with default parameters). We defined high-quality bins as those with completeness >90%, contamination <5%[27] and also passing the chimeric detection implemented in GUNC[36] (version 0.1.1, with default parameters). Medium-quality bins are defined as completeness ≥50% and contamination <10% and low-quality bins are defined as completeness <50% and contamination <10%[27].

**Model transfer between environments**. To evaluate the generalization of the learned models, we selected three models as training sets from the human gut, dog gut, and ocean microbiome datasets. In each dataset, we selected a model from the sample that could generate the highest number, median number, and lowest number of high-quality bins. For the human gut dataset, we termed them human_high, human_median, and human_low, with models from the other environments named analogously. For every environment, we also randomly selected 10 samples from the rest of the samples as testing sets (no overlap between the training sets and testing sets). Then, we transferred these models to the testing sets from the same environment or different environments and used the embeddings from these models to bin the contigs.

To evaluate the effect of training with different numbers of samples, for every environment, we also randomly chose 10 samples as testing sets and trained the model on different numbers of training samples (randomly chosen 1, 3, 5, 10, 15,

and 20 samples; no overlap between the training sets and testing sets). For each number of samples, we randomly chose samples and trained the model 5 times.

To evaluate the pretraining approach, we used another two human gut datasets. We termed SemiBin with a pretrained model from the human gut dataset used before as SemiBin(pretrain; external). We also trained a model from 20 randomly chosen samples from the hold-out datasets and applied it to the same dataset; an approach we termed SemiBin(pretrain; internal). We benchmarked SemiBin(pretrain; external), Metabat2, original SemiBin, and SemiBin(pretrain; internal).

**MAG analyses.** To identify the overlap between the bins from SemiBin and Metabat2 with single-sample binning, we used Mash (version 2.2, with default parameters)[38] to calculate the distance between bins from the two methods. Then, we assigned corresponding bins with Mash distance ≤0.01 and we considered these two bins as the same genome in subsequent comparisons. After obtaining the overlap of bins sets, we classified the high-quality bins from each method into 4 classes: HQ-HQ: also high-quality in the other method; HQ-MQ: medium-quality in the other; HQ-LQ: low-quality or worse in the other; and HQ-Miss: could not be found in the other. We calculated the recall, precision, and F1-score for the HQ-HQ component. Recall and precision are completeness and 1−contamination are estimated from CheckM. F1-score is 2 × (recall × precision)/(recall + precision). To evaluate the species diversity of different methods, we annotated high-quality bins with GTDB-Tk (version 1.4.1, using classify_wf workflow with default parameters)[37].

For the analysis of *B. vulgatus* bins, average nucleotide identity (ANI) comparisons were calculated using fastANI[59] (version 1.32, -fragLen 1000). *B. vulgatus* bins were annotated with Prokka (version 1.14.5, with default parameters)[70]. Pan-genome analyses were carried out using Roary (version 3.13.0, -i 95 -cd 100)[71]. We used Scoary (version 1.6.16, -c BH)[72] to identify genes with significant differences in the human and dog gut microbiome datasets. Phylogeny reconstructions of core genes were performed with IQTREE using 1000 bootstrap pseudoreplicates for model selection (version 1.6.9, -m MFP -bb 1000 -alrt 1000)[73], and visualized with ggtree package (version 1.8.154)[74]. Principal component analysis was done using the prcomp function from the stats[75] package.

**Reporting summary.** Further information on research design is available in the Nature Research Reporting Summary linked to this article.

## Data availability

The sequence data used in the study are publicly available in the ENA with study accessions PRJEB27928, PRJEB20308, PRJEB1787, PRJEB1788, PRJEB4352, PRJEB4419, PRJNA504891, PRJNA290729, PRJEB4391, PRJEB6997, PRJEB7759, PRJEB11755, PRJNA271013, and PRJNA300541. The study accessions the soil dataset see Supplementary Data 2. The simulated CAMI I (low, medium, and high complexity) and CAMI II datasets (skin and oral cavity from Toy Human Microbiome Project Dataset) can be downloaded from https://data.cami-challenge.org/participate. The MAGs generated from real metagenomes in the benchmarking can be obtained from Zenodo: https://doi.org/10.5281/zenodo.5181237 (human gut microbiome MAGs), https://doi.org/10.5281/zenodo.5181385 (dog gut microbiome MAGs), https://doi.org/10.5281/zenodo.5181391(marine microbiome MAGs) and https://doi.org/10.5281/zenodo.5861178 (soil microbiome MAGs). All intermediate results of benchmarking can be found on Github at https://github.com/BigDataBiology/SemiBin_benchmark. Source data are provided with this paper.

## Code availability

Code for the tool[76] can be found on GitHub at https://github.com/BigDataBiology/SemiBin/ and is freely available under the MIT license. The code is also archived on Zenodo at https://doi.org/10.5281/zenodo.4649670 with the version benchmarked in this work being found under https://doi.org/10.5281/zenodo.6006707. The analysis code and intermediate results can be found on Github at https://github.com/BigDataBiology/SemiBin_benchmark (archived on Zenodo under https://doi.org/10.5281/zenodo.6363509).

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

## Acknowledgements

This research was supported by the National Natural Science Foundation of China (grant 31950410544, L.P.C.), the National Natural Science Foundation of China (grant 61932008 and 61772368, X.M.Z.), the Shanghai Municipal Science and Technology Major Project (grant 2018SHZDZX01, X.M.Z., and L.P.C.), the National Key R&D Program of China (grant 2020YFA0712403, X.M.Z.). We thank Anthony Fullam and Thomas Sebastian B. Schmidt (Bork group, European Molecular Biology Laboratory) for providing us with the assembled contigs for the metagenomes used (with the exception of those from PRJNA504891). We thank members of the Coelho and Zhao groups for their comments and discussions throughout the development of the project. We also thank Marija Dmitrijeva (University of Zurich) for helpful comments on a previous version of the manuscript. Beta users of SemiBin are thanked for their suggestions and bug reports.

## Author contributions

L.P.C. and X.Z. conceived the study and supervised the project. S.P., X.Z., and L.P.C. designed the method. S.P. and L.P.C. wrote the software. S.P., C.Z. and L.P.C. designed and performed the analyses. S.P. wrote the first draft of the manuscript. All authors contributed to the revision of the manuscript prior to submission and all authors read and approved the final version.

## Competing interests

All authors declare no competing interests.
