## [Peer Review File · Nature Communications]

REVIEWER COMMENTS

Reviewer #1 (Remarks to the Author):

The manuscript by Pan et al., "SemiBin" describes a new binning tool to obtain microbial genomes directly from metagenomes. SemiBin uses must-link and cannot-link (similar to SolidBin 2019) for contrastive learning: cannot link are obtained from GTDB and must link by breaking longer contigs. Input features are composition and abundance. If less than 5 samples, abundance is used only for clustering. Three modes were tested: single sample, co-assembly (all samples concatenated, all abundances), and multi-sample (all abundances). The neural network generates embeddings for each contig, which are used to calculate a distance matrix between every contig. The contigs are clustered according to that matrix.

The approach was tested on CAMI I and CAMI II and real datasets from the human gut, dog gut, and marine. On CAMI datasets, SemiBin got 6.7% to 65.0% more distinct bins than without embeddings. On CAMI I, SemiBin obtained 11.4-17.4% more HQ bins than the second-best binner. On CAMI II SemiBin got up to 44.3% more HQ bins (only some CAMI II datasets were used - skin and oral because VAMB had its best performance on those). For real data single sample, outperformed MetaBAT2 by 41.2 to 72.0%. For multi-sample (co-abundance), outperformed VAMB 11-30.7%.

In general, I found the evaluation rigorous, and numerous mock and real datasets establish that SemiBin does seem to improve state-of-the-art. The combination of cannot-links and must-links in the binning process is not novel. However, the combination of link-constraints, machine-learning, clustering and re-clustering based on single-copy genes, seems like a significant contribution to the field. I found the comparisons easy to follow and the method nicely outline in the conceptual figure. I only have a few clarifying general comments.

General comments:

- The description of how contigs are chosen to be broken for must-link constraints is a little confusing (L211-213) and seems like an important detail to understand the method. For example, what were the actual minimum sizes, and the effect of having more or less must and cannot links? Furthermore, it seems that the database used to annotate contigs has some effect on these results, but the reason why is not discussed (why is one database better for simulated datasets and another for real datasets).

- While many other parts of the approach is explained and discussed in detail. There is no detailed description of the clustering approach (or clustering approaches). Is it possible to discuss what the effect of the chosen clustering approach is on the results? Would other clustering algorithms work with these embeddings?

- A second clustering step with k-means is also used to break bigger bins, what is the effect of this reclustering step, i.e. how many more HQ bins are obtained? It was not directly visible from Figure 1 and should be included in this overview.

Reviewer #2 (Remarks to the Author):

Pan et al. introduce SemiBin - a new tool to recover genomes from metagenomes. The main conceptual advance is the use of Cannot-link and Must-link edges between contigs in a sample, which is subsequently used to train a neural network. While these concepts were previously introduced in a tool, SolidBin, the implementation by SemiBin appears to be much more robust. Using a variety of simulated and real world datasets, the authors show that SemiBin achieves superior performance compared to existing tools.

Major comments

At the end of the supplement (Table S4) the authors provide running times and RAM requirements for SemiBin. It would be helpful to include these numbers in the main section of the manuscript and include comparisons with existing tools. Please also indicate the cumulative running time and Peak RAM across all steps. Also indicate the number of CPUs used (and not just the EC2 instance type).

Given the high computational requirements of training (table s4), it's important to provide pre-trained models per-biome (assuming there's not a single model that works well across biomes). This will greatly enhance the value of the tool and save the community a lot of duplicated efforts. The authors have already provided models for human, canine, and marine biomes. This is a good start but should be expanded to other commonly analyzed biomes with publicly available data in the SRA. While not necessary, it would be interesting to evaluate how well the models work on different environments - do models trained on human gut work well for marine samples, for example?

It is important to assess if SemiBin's performance depends on the availability of closely related reference genomes in the GTDB. And whether SemiBin's performance is overestimated in the CAMI benchmark due to the presence of CAMI genomes in the database (all CAMI genomes were submitted to NCBI). The authors should benchmark SemiBin on the CAMI metagenomes after excluding genomes from the reference database at different shared taxonomic ranks: species, genus, family, order, class, and phylum. For example, at the species rank, all genomes would be excluded from the GTDB that match the same species present in a CAMI metagenome prior to taxonomic annotation of contigs. This will help to inform readers how well we can expect the tool to perform for community with different levels of novelty and different complexity levels.

For their real-world benchmarking experiments, the authors focused on datasets from human, canine, and marine environments. Each of these environments are well characterized by reference genomes (to varying degrees). It would be valuable to also evaluate the relative performance of SemiBin on soil samples, which represent the most challenging environment for metagenomic assembly and binning.

I think there are many users who will want to apply SemiBin to datasets from diverse environments, but either don't have multiple samples from the same environment (to train a robust model), or don't want to spend the time to train a model for each environment type. In some cases, the user may not even know the environment type. Is there a way for the authors to pre-train a model that works well in a variety of different environment types?

If not, why is this the case? Why would a model work well in one environment, but not for another? I think this is an interesting question worth addressing in the paper. Is it largely due to differences in diversity? For example, the model may be very strict in one environment where there are closely related taxa, and more lenient in another where the taxa are less similar? If this is the case, I wonder if it would be possible to select a pre-trained model for the user based on the estimated community diversity (or another metric).

Minor comments

How do the authors plan to keep their MMseqs2 database in sync with new versions of the GTDB? Is it possible to provide a command to perform the update?

Fig 1b. Typo: Extra space after "contig"

Fig 1e: Typo: should read "weighted" k-means

Line 127: grammatical error "if more number of samples"

Reviewer #3 (Remarks to the Author):

- This is an interesting manuscript that uses a particular architecture of deep neural networks semi-supervised with taxonomic annotation to provide improved bins of metagenomic sequences. The approach is interesting and the experimental design is extensive. However, there are a number of questions and issues, that once properly answered could improve the manuscript. The article is well written, although the narrative can be improved as it is not always easy for the reader to follow and needs to find information back and forth to understand the concepts.

- The binning is quite important in the microbiome field as it allows to bring structure to the unknown world of the microbiome, where the high throughput sequencing process breaks the structure of the DNA. In the introduction and Figure 1 it would be interesting to remind the reader, where one goes from binning, as the quality of the bins depend on the types of applications that will follow. For instance, reconstructing genomes needs to have as many reads as possible, but for quantitative applications it is the coverage of part of the bin that will be important (Nielsen et al. Nat Biotech 2014).

- By the way this is an important historical/methodological paper in the binning field introducing a co-abundance approach and is not cited. One can also wonder, how SemiBin compares to this approach? Other derivatives of this approach have since appeared such as MSPminer <https://doi.org/10.1093/bioinformatics/bty830> aiming to increase bin size.

- Is the binning performed at the contig level or also at the read level ? It is not clear how they go from one concept to the other. They even mention genes at some point in Figure 1. Can they clarify and check coherence throughout the manuscript ?

- It may be helpful for the reader's understanding to introduce early on how are the embeddings for the data calculated ? Illustrate and describe in Figure 1.

- What is the impact of sequencing depth in the bin quality ?

- After binning and alignment, how is the distribution of abundance throughout known reference genomes ? line 230: The authors make the assumption that the number of reads per base obeys normal distribution. Do they have any evidence which validates this assumption ?

- What is the impact of the constraints (cannot-link and must-link) on the results, what percentage of contigs have these constraints ? What is the impact of the contig annotations on the results ? Can the authors quantify these ? It is important to show which parameters improve the binning as they are stacked together in the overall method along with deep learning, embeddings, etc?
- line 18: ... retaining the capability of binning genomes => should be binning reads/sequences/contigs not genomes as this is not demonstrated.
- line 53: must-link constraints are generated by breaking up longer contigs. Does the contig size have an influence on the results. It would be interesting for the reader to see some range of proportions of contigs used for this purpose.
- line 55: caution on the terms used. The embedding is a transformation of the input data by the deep neural network model. The embedding is not a model per se. This term is used often and it is not clear for the reader how these embeddings are extracted. On which part of the network specifically ? Have different embeddings been tested and what is their impact on the results? What is the dimension of the embedding space ?
- How is contig abundance and kmer frequency input precisely in the network ? It may be useful to improve Figure 1 to take illustrate this or build a specific figure/panel.
- Figure 2, the name of the methods in the figure are cryptic, why not using the full names as introduced in the text and legend ? Please check the correspondence of text and figures/legends in their the methods are called. They should be the same.
- Figure 3, legend, please specify the meaning of the axes especially the b right panel
- Have the authors tried to use bin assembly as a performance indicator ? This may bring more evidence on the performance of the method.
- In which dataset the model (siamese network) was trained? And on which it was tested. The experimental ML design is not very clear
- line 132, for each environment? The pre-trained model was trained in how many samples ? And tested in how many others ? Not clear...
- line 237: the phrase To remove... is confusing. Could the authors reformulate /clarify?
- How does the max_edges parameter affect the number of contigs left in the graph and subsequently the size of the bins ? It is not clear what the impact of this parameter is.
- How are the cannot-link and must-link constrains coded and input in the network ?

Response to the reviewers

We thank the reviewers for their critical assessment of our work. Below, we address their concerns point by point, but at a high level, the largest changes with respect to the previous submission are:

1. We now add a soil benchmark, where SemiBin also outperformed the alternatives.
2. We now include pretrained models for a total of 10 different habitats as well as a generic model. The results of thoroughly evaluating them are presented in a new Figure (Fig. 5).
3. We tested variations in parameter settings (and even different algorithms for the final clustering and reclustering steps) and show that the SemiBin architecture is robust to these choices.

Reviewer 1

The manuscript by Pan et al., "SemiBin" describes a new binning tool to obtain microbial genomes directly from metagenomes. SemiBin uses must-link and cannot-link (similar to SolidBin 2019) for contrastive learning: cannot link are obtained from GTDB and must link by breaking longer contigs. Input features are composition and abundance. If less than 5 samples, abundance is used only for clustering. Three modes were tested: single sample, co-assembly (all samples concatenated, all abundances), and multi-sample (all abundances). The neural network generates embeddings for each contig, which are used to calculate a distance matrix between every contig. The contigs are clustered according to that matrix.

The approach was tested on CAMI I and CAMI II and real datasets from the human gut, dog gut, and marine. On CAMI datasets, SemiBin got 6.7% to 65.0% more distinct bins than without embeddings. On CAMI I, SemiBin obtained 11.4-17.4% more HQ bins than the second-best binner. On CAMI II SemiBin got up to 44.3% more HQ bins (only some CAMI II datasets were used - skin and oral because VAMB had its best performance on those). For real data single sample, outperformed MetaBAT2 by 41.2 to 72.0%. For multi-sample (co-abundance), outperformed VAMB 11-30.7%.

In general, I found the evaluation rigorous, and numerous mock and real datasets establish that SemiBin does seem to improve state-of-the-art. The combination of cannot-links and must-links in the binning process is not novel. However, the combination of link-constraints, machine-learning, clustering and re-clustering based on single-copy genes, seems like a significant contribution to the field. I found the comparisons easy to follow and the method nicely outline in the conceptual figure. I only have a few clarifying general comments.

Comment 1.1 — The description of how contigs are chosen to be broken for must-link constraints is a little confusing (L211-213) and seems like an important detail to understand the method. For example, what were the actual minimum sizes, and the effect of having more or less must and cannot links?

Response: By default, SemiBin uses a heuristic method to automatically find the minimum size of contigs to break up: SemiBin chooses a minimum size that makes contigs as long as or longer than this threshold can contain most (98%) basepairs of the input contigs. If this minimum size smaller than 4,000bps, the minimum will be

set as 4,000bps. However, in most of the situations of the benchmarking, especially real datasets, the minimum threshold is 4,000bps.

We tested different minimum size threshold ($> 1,000\text{bps}$, $> 4,000\text{bps}$, and $> 10,000\text{bps}$) for generating *must-link* constraints and different number (10,000, 500,000, 4,000,000, and 10,000,000) of *cannot-link* constraints. We compared the results (number of high-quality bins) in these situations (see Supplementary Fig. 20). SemiBin is robust to these parameter variations, except when the number of *cannot-link* constraints and the threshold for generating *must-link* constraints are too small. Because breaking up short contigs will lead to noise (from short contigs) and small number of *cannot-link* constraints can not provide enough information for the training. The number of negative samples is important in the contrastive learning¹.

Changes made: We updated the description of how contigs are chosen to be broken for *must-link* constraints, (see L252-L257). We added the effect of different number of *must-link* and *cannot-link* constraints to the binning results in the Supplementary Text (see L483-L493) and Supplementary Fig. 20.

Comment 1.2 — Furthermore, it seems that the database used to annotate contigs has some effect on these results, but the reason why is not discussed (why is one database better for simulated datasets and another for real datasets).

Response: On the CAMI I datasets, we found that the annotation from CAT+NCBI is better than mm-seqs2+GTDB (more accurate *cannot-link* constraints and these *cannot-link* constraints can cover more genomes in the dataset, see Supplementary Table 4). We speculate that CAMI I datasets are well covered by NCBI, while real datasets may benefit more from the added coverage provided by GTDB. In the absence of ground truth, it is impossible to be certain, but we would expect that this would lead to more accurate *cannot-link* constraints.

Changes made: We added Supplementary Table 4, quantifying the accuracy of links derived from annotations (in simulated data).

Comment 1.3 — While many other parts of the approach is explained and discussed in detail. There is no detailed description of the clustering approach (or clustering approaches). Is it possible to discuss what the effect of the chosen clustering approach is on the results? Would other clustering algorithms work with these embeddings?

Response: In SemiBin, we converted the embeddings to a graph and used the Infomap algorithm which was already used in other binning tool (bin3C²) to detect the communities (bins).

We now tested a few alternatives, namely Label propagation³, Leiden⁴ and Louvain⁵ community detection algorithms (these are the ones available from the igraph package⁶) to show the results on the CAMI I and II datasets (see Supplementary Fig. 23). We can see that besides the Leiden algorithm, other method showed similar results in the CAMI I and II datasets, indicating that the embeddings can be robust to some other community detection methods (seems Leiden is not suitable for this question).

For the reclustering step, we also tried several alternative clustering methods besides weighted *k*-means, namely SpectralClustering⁷, AgglomerativeClustering and DBSCAN⁸ (see Supplementary Fig. 24). In these tests, weighted *k*-means outperformed other alternatives.

Changes made: We described the analysis of different clustering methods in the Supplementary Text (see L507-L517). We showed the results in Supplementary Fig. 23 and 24.

Comment 1.4 — A second clustering step with k-means is also used to break bigger bins, what is the effect of this reclustering step, i.e. how many more HQ bins are obtained? It was not directly visible from Figure 1 and should be included in this overview.

Response: We agree with the reviewer that the previous version of the manuscript did not provide sufficient information on this question.

In the simulated datasets, the reclustering step can generate significantly more high-quality bins (22.7%, 72.5% and 258.8% in CAMI I datasets and 159.2% and 50.5% in CAMI II datasets, see Supplementary Fig. 26). In the human gut, dog gut, ocean and soil real datasets, there are 30/1700 (1.76%), 1/2797 (0.04%), 9/444 (2.03%), 4/103 (3.88%) with single-sample binning and 105/1549 (6.78%), 216/3448 (6.26%), 44/570 (7.72%), 3/209 (1.44%) high-quality bins coming from reclustering.

In the real datasets, the improvement is marginal, perhaps due to the more complex environment compared to the simulated datasets. Nonetheless, as there are some gains and we were able to reduce the performance costs in the newer SemiBin releases, reclustering is now the default.

Changes made: We described the influence of the reclustering step in the Supplementary Text (see L549-L554). We showed the results in the Supplementary Fig. 26. We added the reclustering to Fig. 1 and the description of reclustering in the main text (see L72-L73). We improved the performance of the reclustering step and made it the default.

Reviewer 2

Pan et al. introduce SemiBin - a new tool to recover genomes from metagenomes. The main conceptual advance is the use of Cannot-link and Must-link edges between contigs in a sample, which is subsequently used to train a neural network. While these concepts were previously introduced in a tool, SolidBin, the implementation by SemiBin appears to be much more robust. Using a variety of simulated and real world datasets, the authors show that SemiBin achieves superior performance compared to existing tools.

Comment 2.1 — At the end of the supplement (Table S4) the authors provide running times and RAM requirements for SemiBin. It would be helpful to include these numbers in the main section of the manuscript and include comparisons with existing tools. Please also indicate the cumulative running time and Peak RAM across all steps. Also indicate the number of CPUs used (and not just the EC2 instance type).

Response: We agree with the reviewer that cumulative time and peak RAM are very important for users to evaluate whether the tool is appropriate for their use case. Note too that we were able to make several improvements to performance (generating data and binning step, especially relevant when using a pretrained model), and updated the benchmarks to reflect this (see Supplementary Table 5 and Table 6).

Changes made: We added the following information to the Methods (see L387-L392) and the caption of the Supplementary Table 5 and Table 6: "CPU machine: 1 CPU;8 physical core(cores) ; 16 logical cores(processors)."

We added more description of the running time and memory usage in the main text (see L138-L147). We changed the average memory usage to peak memory usage in Supplementary Table 5 and added comparisons to existing tools of the cumulative running time and peak memory usage in Supplementary Table 6.

Comment 2.2 — Given the high computational requirements of training (table s4), it's important to provide pre-trained models per-biome (assuming there's not a single model that works well across biomes). This will greatly enhance the value of the tool and save the community a lot of duplicated efforts. The authors have already provided models for human, canine, and marine biomes. This is a good start but should be expanded to other commonly analyzed biomes with publicly available data in the SRA. While not necessary, it would be interesting to evaluate how well the models work on different environments - do models trained on human gut work well for marine samples, for example?

Response: We whole-heartedly agree with the reviewer that expanding the set of habitats is a strong addition to the tool. We had considered this future work, but motivated by the reviewer's comment, we have moved it forward and SemiBin now (since v0.6, released on Feb 8) includes pretrained models for a total of 10 habitats (human gut, dog gut, ocean, soil, cat gut, human oral, mouse gut, pig gut, built environment and wastewater). These are all based on GMGCv1 (Global Microbial Gene Catalog, version 1,⁹), which contains 14 habitats; but 4 of them did not contain enough deeply sequenced samples. In all environments, SemiBin could generate 11.8%–240.4% more high-quality bins than Metabat2 except wastewater, whose assembled contigs were very few and both tools were hard to extract bins from this environment (see Fig. 5).

Also, we added empirical results from the the cross validation of the ten pretrained models (see Fig. 5). Indeed, habitat-specific models do appear to perform better than a generic one in most of the situations. When using model to the different environment from the training environment, the model can also get good results, and in most these situations, SemiBin can still perform better than Metabat2.

Changes made: Starting in version 0.6 of SemiBin, we now provide pre-built models for 10 habitats in GMGCv1 (Coelho et al., 2022): human gut, dog gut, and marine (as before); as well as soil, cat gut, mouse gut, pig gut, built environment, human oral, and wastewater. We described the results of the pretrained model in the main text (see L181-L187). We showed the results of the pretrained model and the cross validation in the Fig. 5.

Comment 2.3 — It is important to assess if SemiBin's performance depends on the availability of closely related reference genomes in the GTDB. And whether SemiBin's performance is overestimated in the CAMI benchmark due to the presence of CAMI genomes in the database (all CAMI genomes were submitted to NCBI). The authors should benchmark SemiBin on the CAMI metagenomes after excluding genomes from the reference database at different shared taxonomic ranks: species, genus, family, order, class, and phylum. For example, at the species rank, all genomes would be excluded from the GTDB that match the same species present in a CAMI metagenome prior to taxonomic annotation of contigs. This will help to inform readers how well we can expect the tool to perform for community with different levels of novelty and different complexity levels.

Response: Like the reviewer, we were under the impression that all CAMI genomes had been submitted to NCBI. However, after corresponding with the CAMI authors, we realized that we were mistaken and they are not a part of the database (due to quality issues).

Nonetheless, it is possible to remove matching genomes from the database at different taxonomic ranks (see Supplementary Text). The results (see Supplementary Fig. 27) showed that SemiBin did not rely on the closely related genomes in the reference databases, and only a very minor loss in the number of high-quality bins recovered, even when excluding all genomes that matched the same phylum from the GTDB databases.

Changes made: We described how to remove matched genomes and the results in the Supplementary Text (see L566-L575). We showed the results in Supplementary Fig. 27.

Comment 2.4 — For their real-world benchmarking experiments, the authors focused on datasets from human, canine, and marine environments. Each of these environments are well characterized by reference genomes (to varying degrees). It would be valuable to also evaluate the relative performance of SemiBin on soil samples, which represent the most challenging environment for metagenomic assembly and binning.

Response: We agree and have added results from soil (a project with 101 samples) throughout the main text, including Figure 3. This was, indeed, a challenging environment and none of the tools returns a high number of high quality bins. However, in relative terms, SemiBin performs the best. SemiBin(pretrain) can reconstruct 44 (74.6%) more high-quality bins compared to Metabat2 with single-sample binning and SemiBin can reconstruct 132 (171.4%) more high-quality bins compared to VAMB with multi-sample binning (see Fig. 3).

Changes made: We described the results of soil dataset in the main text (see L127-173). We added the results of soil dataset to the Fig. 3, Supplementary Fig. 10, 11, 13, 14 and 15.

Comment 2.5 — I think there are many users who will want to apply SemiBin to datasets from diverse environments, but either don't have multiple samples from the same environment (to train a robust model), or don't want to spend the time to train a model for each environment type. In some cases, the user may not even know the environment type. Is there a way for the authors to pre-train a model that works well in a variety of different environment types?

Response: We agree that this would be a strong addition. We attempted to learn a model (which we termed as SemiBin(global)) from all habitats. The results show that it is possible to build a model that performs better than Metabat2 across a large range of habitats. In each habitat, though, a habitat-specific model still generally returns more high-quality bins. Thus, we recommend that users choose a habitat-specific model if possible, and use SemiBin(global) as a fallback.

Changes made: We added a pretrained model training from 10 environments to SemiBin. We described the results of SemiBin(global) in the main text (see L188-L191). We show the results of SemiBin(global) in Fig. 5.

Comment 2.6 — If not, why is this the case? Why would a model work well in one environment, but not for another? I think this is an interesting question worth addressing in the paper.

Response: Thanks for these comments that are very meaningful for our work. First, as mentioned above, we now provide a model (termed *SemiBin(global)*) that was trained from all environments. We show it performs well across all environments (outperformed Metabat2 in all situations, with small costs losses compared to *SemiBin(pretrain)*). We think that if the users do not know what the environment is, *SemiBin(global)* is a good solution.

As we now more thoroughly evaluated model transfer, we could observe some basic patterns. For example, the best models for the human gut (besides the human gut itself) are other mammalian guts; except for the mouse gut, which we (and others) had already reported as more different from the human gut in composition than other habitats.

Changes made: We added a new figure (Fig. 5) showing the effect of transferring a model trained in one habitat to another and briefly discuss the effects observed in the Discussion (L219-L224).

Comment 2.7 — Is it largely due to differences in diversity? For example, the model may be very strict in one environment where there are closely related taxa, and more lenient in another where the taxa are less similar? If this is the case, I wonder if it would be possible to select a pre-trained model for the user based on the estimated community diversity (or another metric).

Response: As the reviewer, we speculated that factors such as the presence of closely related taxa may be driving differences in performance between habitats. Indeed, as shown in the new Fig. 5, models from *similar* environments perform better than ones from very different ones. However, we did not find that this was simply a matter of community diversity (see R. Fig. 1).

Response Figure 1. Neither estimated species or unigene richness (after downsampling to 1 million reads) is predictive of SemiBin model fit. Each point represents a pair of habitats, plotted according to their estimated species (**left**) or unigene (**right**) richness and how well a model performs when transferred from one habitat to the other (relative to the model trained on the same habitat). Estimates of species/unigene richness were performed as in *Coelho et al., 2022*⁹; while the estimate of model fit was performed as in Fig. 5 of the main text. The Spearman correlations are not statistically significant ($P=0.96$ and $P=0.68$, for species and unigene, respectively). Using Shannon entropy as an alternative diversity metric similarly failed to return a statistically significant correlation.

Minor

Comment 2.8 — How do the authors plan to keep their MMseqs2 database in sync with new versions of the GTDB? Is it possible to provide a command to perform the update?

Response: This is a good question. In fact, other users have asked similar questions and requested similar features.

For now, we have chosen to provide a particular version of GTDB for user convenience and reproducibility; but SemiBin is not dependent on it. This is now clearer in the online documentation, including step-by-step instructions on how to update to the latest version of GTDB (one can use mmseqs-internal tools for it and, subsequently, point SemiBin to them).

In the latest release of SemiBin, we also added the ability for the user to provide a contig classification (which should be in the same file format that mmseqs uses with GTDB-formatted identifiers), but any version of GTDB

can be used.

Changes made: We updated the online documentation: <https://semibin.readthedocs.io/en/latest/faq/>

Comment 2.9 — Fig 1b: Typo: Extra space after "contig"

Changes made: We have fixed it. See Fig. 1.

Comment 2.10 — Fig 1e: Typo: should read "weighted" k-means

Changes made: We have fixed it. See Fig. 1.

Comment 2.11 — Line 127: grammatical error "if more number of samples"

Changes made: We have changed it to 'if a large enough number of samples is used in pretraining'. See L148.

Reviewer 3

This is an interesting manuscript that uses a particular architecture of deep neural networks semi-supervised with taxonomic annotation to provide improved bins of metagenomic sequences. The approach is interesting and the experimental design is extensive. However, there are a number of questions and issues, that once properly answered could improve the manuscript. The article is well written, although the narrative can be improved as it is not always easy for the reader to follow and needs to find information back and forth to understand the concepts.

Comment 3.1 — The binning is quite important in the microbiome field as it allows to bring structure to the unknown world of the microbiome, where the high throughput sequencing process breaks the structure of the DNA. In the introduction and Figure 1 it would be interesting to remind the reader, where one goes from binning, as the quality of the bins depend on the types of applications that will follow. For instance, reconstructing genomes needs to have as many reads as possible, but for quantitative applications it is the coverage of part of the bin that will be important (Nielsen et al. Nat Biotech 2014).

Changes made: Thanks for this comment. We have added more description of the application of metagenomic binning in the introduction section (see L28-L31, L56-L59).

Comment 3.2 — By the way this is an important historical/methodological paper in the binning field introducing a co-abundance approach and is not cited. One can also wonder, how SemiBin compares to this approach? Other derivatives of this approach have since appeared such as MSPminer <https://doi.org/10.1093/bioinformatics/bty> aiming to increase bin size.

Response: We agree that Canopy (Nielsen et al. Nat Biotech 2014) is an important method and foundational paper in the field (and now cite it). However, as we now make clearer, Canopy and MSPminer are solving slightly different problems. Canopy bins co-abundant genes to get metagenomic species (MGS) first and use a

MGS-augmented genome assembly method (map reads to MGS genes and then assemble MGS-specific reads) to get the bins¹¹. MSPminer extended this approach, which resulted in capturing only core genes from a species, and returns Metagenomic Species Pan-genomes (MSPs)¹². This is related to the problem of binning contigs that SemiBin solves, but it is not directly comparable.

Note that VAMB (which we included already in the first submission) from the same group as (Nielsen et al., 2014) is, in many ways the descendant of the co-abundance approach pioneered by Canopy. In the VAMB paper, they demonstrated that VAMB significantly outperformed Canopy in the CAMI II datasets (the same dataset used in our manuscript).

For this reason and because the problems they solve are different (it does not even make sense to evaluate how well these tools work on each sample as they operate on large datasets as a whole), we do not include Canopy and MSPminer in the benchmark.

Changes made: In the introduction, we briefly mention the approach used by Canopy and MSPminer and differentiate ours from it (see L32-L37).

Comment 3.3 — Is the binning performed at the contig level or also at the read level ? It is not clear how they go from one concept to the other. They even mention genes at some point in Figure 1. Can they clarify and check coherence throughout the manuscript ?

Response: The binning is performed at the contig level. We have described the difference of binning contigs and binning gene catalogs and clarified the binning at the contig level in the introduction section, Fig. 1 and the results section.

Changes made: We have added the clarification of the binning at contig level in the introduction section (see L36-L37), Fig. 1 and the results section (see L62-L63).

Comment 3.4 — It may be helpful for the reader's understanding to introduce early on how are the embeddings for the data calculated ? Illustrate and describe in Figure 1.

Changes made: We have updated the Fig. 1 to show how the embeddings are calculated from the neural network (the output layer, see Fig. 1). We also updated the Supplementary Fig. 2 to show how the features are input to the model.

Comment 3.5 — What is the impact of sequencing depth in the bin quality ?

Response: As expected, there is a positive relationship between the number of HQ bins extracted and the sequencing depth, but as shown in Response Fig. 2, this is only a weak relationship.

Response Figure 2. There is a modest positive correlation between the sequencing depth and the number of extracted bins. There is a generally positive correlation between sequencing depth and number of bins extracted, but the strength is modest and observed mainly at relatively low depths.

Comment 3.6 — After binning and alignment, how is the distribution of abundance throughout known reference genomes ? line 230: The authors make the assumption that the number of reads per base obeys normal distribution. Do they have any evidence which validates this assumption ?

Response: This was introduced by MetaBAT (Kang et al., 2015), where it is shown empirically that this is a good approximation (see Fig 2E in that manuscript).

Changes made: We added a citation to Kang et al., 2015 in L234.

Comment 3.7 — What is the impact of the constraints (cannot-link and must-link) on the results, what percentage of contigs have these constraints ? What is the impact of the contig annotations on the results ? Can the authors quantify these ? It is important to show which parameters improve the binning as they are stacked together in the overall method along with deep learning, embeddings, etc?

Response: We tested different minimum size threshold (1,000bps, 4,000bps, and 10,000bps) for generating *must-link* constraints and different number (10,000, 500,000, 4,000,000, and 10,000,000) of *cannot-link* constraints. We compared the number of high-quality bins in these situations. SemiBin is robust to these parameter variations,

except when the number of *cannot-link* constraints and the threshold for generating *must-link* constraints are too small (see Supplementary Fig. 20).

To test the influence of the contig annotations on the results, we have tried to remove matching genomes to the CAMI datasets from the GTDB reference genomes at different taxonomic ranks and then annotated the contigs with corresponding databases (see Supplementary Text). The results (see Supplementary Fig. 27) showed that SemiBin did not rely on the closely related genomes in the reference databases and only a very minor loss in the number of high-quality bins recovered, even when excluding all genomes that matched the same phylum from the GTDB databases. It indicated that SemiBin can be robust to the contig annotations from different reference genomes.

From our benchmarking (see the influence of clustering and reclustering part from Supplementary Fig. 23, 24 and 26), the superior results are from the all parts stacked together (deep learning embedding, clustering and reclustering).

Changes made: We added the effect of different number of *must-link* and *cannot-link* constraints to the binning results in the Supplementary Text (see L483-L493) and Supplementary Fig. 20. We added the percentage of contig have the *must-link* and *cannot-link* constraints in the Supplementary Table 7. We showed the influence of different contig annotations by removing matched genomes from the reference genomes in Supplementary Fig. 27 and Supplementary Text (see L566-L575).

Comment 3.8 — line 18: ... retaining the capability of binning genomes => should be binning reads/sequences/contigs not genomes as this is not demonstrated.

Response & changes made: We agree that the sentence was imprecise and changed this to read 'SemiBin exploits the information in reference genomes, while retaining the capability of reconstructing high-quality bins that are outside the reference dataset.' (see L18-L20)

Comment 3.9 — line 53: *must-link* constraints are generated by breaking up longer contigs. Does the contig size have an influence on the results. It would be interesting for the reader to see some range of proportions of contigs used for this purpose.

Response: SemiBin, as is typical for binning tools, filters out short contigs (by default, those shorter than 2,500 bps). The main reason is that, for computing features such as kmer, short sequences do not provide reliable estimates. Therefore, when we break up a contig, it is desirable that both halves are long enough to provide reliable features.

We have now explored the influence of minimum contig size threshold (> 1,000bps, > 4,000bps, and > 10,000bps) for generating *must-link* constraints and different number (10,000, 500,000, 4,000,000, and 10,000,000) of *cannot-link* constraints. We compared the results (number of high-quality bins) in these situations (see Supplementary Fig. 20). Overall, SemiBin is robust to the changing this minimum contig size threshold, except when the number of *cannot-link* constraints and the threshold for generating *must-link* constraints are too small.

Changes made: We added the effect of different number of *must-link* and *cannot-link* constraints to the binning results in the Supplementary Text (see L483-L493) and Supplementary Fig. 20.

Comment 3.10 — line 55: caution on the terms used. The embedding is a transformation of the input data by the deep neural network model. The embedding is not a model per se.

Changes made: We changed the sentence to read “A deep siamese neural network (*i.e.*, a network consisting of two identical subnetworks, so that pairs of inputs are processed in parallel) is then used to learn a model which transforms the original k -mer and abundance features into a transformed space (the embedded features, see Methods).” (see L68-L70)

Comment 3.11 — This term is used often and it is not clear for the reader how these embeddings are extracted. On which part of the network specifically? Have different embeddings been tested and what is their impact on the results? What is the dimension of the embedding space?

Response: The embeddings we used by default are the features (100 dimensions) from the output layer of the encoder of the semi-supervised siamese neural network (see Supplementary Fig. 2). We now also benchmarked using output from the first (512 dimensions) or the second (512 dimensions) hidden layers (see Supplementary Fig. 19) for the binning. Features from the second hidden layer and the output layer showed similar results, while the first layer resulted in worse performance.

Changes made: We described more clearly of the which layer we use to get the embeddings and what the dimension is in the Methods (see L272-L274). We described the results of different embeddings in Supplementary Fig. 19.

Comment 3.12 — How is contig abundance and kmer frequency input precisely in the network? It may be useful to improve Figure 1 to take illustrate this or build a specific figure/panel.

Changes made: We have updated the Supplementary Fig. 2 to show how the k -mer frequencies and the abundance features are input to the model (see Supplementary Fig. 2).

Comment 3.13 — Figure 2, the name of the methods in the figure are cryptic, why not using the full names as introduced in the text and legend? Please check the correspondence of text and figures/legends in their the methods are called. They should be the same.

Changes made: We have updated the figure so that the names are spelled out.

Comment 3.14 — Figure 3, legend, please specify the meaning of the axes especially the b right panel

Changes made: For the b right panel, what we want to show is the number of extra high-quality bins SemiBin can get compared to other bidders per sample. We have updated the Fig. 3 to clarify the meaning of the axes (see Fig. 3).

Comment 3.15 — Have the authors tried to use bin assembly as a performance indicator? This may bring more evidence on the performance of the method.

Response: We fear we may have misunderstood the reviewer's comment, but our best understanding was that this referred to the fraction of the assembly that can be binned as high-quality bins. We thus computed this metric and confirmed that SemiBin performs better than the other alternatives (see Supplementary Fig. 29).

Changes made: We showed the results of the fraction of the assembly that can be binned as high-quality bins in the Supplementary Fig. 29.

Comment 3.16 — In which dataset the model (siamese network) was trained? And on which it was tested. The experimental ML design is not very clear

Response: By default, because binning is a clustering task, SemiBin is trained on one sample to get the model and this same model is applied to the same sample to obtain the binning results. For SemiBin(pretrain), we trained the model from several samples and applied the pretrained model to the whole project (see Fig. 3). We also used two hold-out projects to show the generalization of the pretrained model (see Supplementary Fig. 16). For the remaining cases, we used 10 samples as the testing set to evaluate the pretrained model, there was no overlap of the training and testing samples (see Figs. 3a, 5 and Supplementary Figs. 12).

Changes made: We clarified the ML design in the Methods section (see L382-L386).

Comment 3.17 — line 132, for each environment? The pre-trained model was trained in how many samples ? And tested in how many others ? Not clear...

Response: For each environment in Fig. 3 (human gut, dog gut, ocean and soil), we used the number of samples that performed best for pretraining (for human gut, dog gut and ocean datasets, model trained from 20 samples and for soil dataset, model trained from 15 samples; see Fig. 3a) and tested the pretrained model in the whole project. We also used two hold-out projects to show the generalization of the pretrained model (see Supplementary Fig. 16).

Changes made: We updated the sentence to make it clearer (see L154-L156) and clarified the ML design in the Methods section (see L382-L386).

Comment 3.18 — line 237: the phrase To remove... is confusing. Could the authors reformulate /clarify?

Changes made: We have clarified this sentence (see L282-L285).

Comment 3.19 — How does the max_edges parameter affect the number of contigs left in the graph and subsequently the size of the bins ? It is not clear what the impact of this parameter is.

Response: The *max_edges* parameter controls the number of edges of each node (each contig) in the graph that will be considered in binning. So it does not affect the number of contigs left in the graph and the size of the bins, only the connections of the contigs (edges). To show the influence of the changes in the *max-edges* parameter to the binning results, we benchmarked different *max-edges* parameter (200, 500, 1000) on the CAMI I (see Supplementary Fig. 21), CAMI II and real datasets (see Supplementary Fig. 22). In the simulated and ocean

datasets, the final binning results were robust to different *max-edges* parameters (slight influence in the ocean datasets). In the human and dog gut dataset, the results were worse when the *max-edges* was larger. It indicated that the larger *max-edges* would introduce contamination to the final results when there were too many edges in the generated graph. Another impact of the *max-edges* parameter was that when this parameter was bigger, more edges would be in the graph, so it need more time for the community detection. Considering the results and computation cost, we set *max-edges* parameter as 200 for the default setting.

Changes made: We described the influence of *max-edges* parameter in the CAMI I, CAMI II and real datasets in the Supplementary Text (see L494-L506). We showed the results of the influence of *max-edges* parameter in the CAMI II and real datasets in Supplementary Fig. 22.

Comment 3.20 — How are the cannot-link and must-link constraints coded and input in the network ?

Response: They are not input directly to the network, but used in the loss function used for training. In the loss function, we term the *must-link* constraint as the positive label and the *cannot-link* constraint as the negative label (see Methods).

Changes made: We updated the description of the loss function in the Methods section to make the usage of the *must-link* and *cannot-link* more clearly (see L260-L261).

References

1. Chen, T., Kornblith, S., Norouzi, M. & Hinton, G. A simple framework for contrastive learning of visual representations. In *International conference on machine learning*, 1597–1607 (PMLR, 2020).
2. DeMaere, M. Z. & Darling, A. E. bin3c: exploiting hi-c sequencing data to accurately resolve metagenome-assembled genomes. *Genome biology* **20**, 1–16 (2019).
3. Raghavan, U. N., Albert, R. & Kumara, S. Near linear time algorithm to detect community structures in large-scale networks. *Phys. review E* **76**, 036106 (2007).
4. Traag, V. A., Waltman, L. & Van Eck, N. J. From louvain to leiden: guaranteeing well-connected communities. *Sci. reports* **9**, 1–12 (2019).
5. Blondel, V. D., Guillaume, J.-L., Lambiotte, R. & Lefebvre, E. Fast unfolding of communities in large networks. *J. statistical mechanics: theory experiment* **2008**, P10008 (2008).
6. Csardi, G., Nepusz, T. *et al.* The igraph software package for complex network research. *InterJournal, complex systems* **1695**, 1–9 (2006).
7. Shi, J. & Malik, J. Normalized cuts and image segmentation. *IEEE Transactions on pattern analysis machine intelligence* **22**, 888–905 (2000).
8. Ester, M., Kriegel, H.-P., Sander, J., Xu, X. *et al.* A density-based algorithm for discovering clusters in large spatial databases with noise. In *kdd*, vol. 96, 226–231 (1996).
9. Coelho, L. P. *et al.* Towards the biogeography of prokaryotic genes. *Nature* 1–5 (2021).

10. Ondov, B. D. *et al.* Mash: fast genome and metagenome distance estimation using minhash. *Genome biology* **17**, 1–14 (2016).
11. Nielsen, H. B. *et al.* Identification and assembly of genomes and genetic elements in complex metagenomic samples without using reference genomes. *Nat. biotechnology* **32**, 822–828 (2014).
12. Plaza Oñate, F. *et al.* Mspminer: abundance-based reconstitution of microbial pan-genomes from shotgun metagenomic data. *Bioinformatics* **35**, 1544–1552 (2019).

REVIEWERS' COMMENTS

Reviewer #1 (Remarks to the Author):

The authors have made a great effort with the responses and the updated manuscript. I have no further comments at this point.

Reviewer #2 (Remarks to the Author):

The authors have addressed all of my points - thank you.

Reviewer #3 (Remarks to the Author):

I thank the authors for their work and for providing point by point answers to the questions, while performing additional experiments and improving the manuscript.